# Does One-shot Give the Best Shot? Mitigating Model Inconsistency in One-shot Federated Learning

Hui Zeng [* 1]  Wenke Huang [* 2]  Tongqing Zhou [1]  Xinyi Wu [1]  Guancheng Wan [2]  Yingwen Chen [1]  Zhiping Cai [1]

## Abstract

Turning the multi-round vanilla Federated Learning into one-shot FL (OFL) significantly reduces the communication burden and makes a big leap toward practical deployment. However, this work empirically and theoretically unravels that existing OFL falls into a garbage (inconsistent one-shot local models) in and garbage (degraded global model) out pitfall. The inconsistency manifests as divergent feature representations and sample predictions. This work presents a novel OFL framework FAFI that enhances the one-shot training on the client side to essentially overcome inferior local uploading. Specifically, unsupervised feature alignment and category-wise prototype learning are adopted for clients' local training to be consistent in representing local samples. On this basis, FAFI uses informativeness-aware feature fusion and prototype aggregation for global inference. Extensive experiments on three datasets demonstrate the effectiveness of FAFI, which facilitates superior performance compared with 11 OFL baselines (+10.86% accuracy). Code available at `https://github.com/zenghui9977/FAFI_ICML25`

## 1. Introduction

As a distributed machine learning paradigm featured with privacy-preserving, Federated Learning (FL) enables multiple clients to collaboratively integrate their knowledge without exposing their local data (McMahan et al., 2017; Zeng et al., 2021; 2025; Karimireddy et al., 2020; Yurochkin et al., 2019; Tang et al., 2024b; Huang et al., 2024; Wan

et al., 2025). Basically, FL allows clients to train local models independently, collects the locally trained models for aggregation, and broadcasts the aggregated global model for iterative local training. However, such a multi-round client-server interaction process would incur a heavy communication burden (e.g., more than 250GB for a simple VGG19 model cumulatively (Zeng et al., 2024; Wu et al., 2020)) and high communication time (e.g., more than 194 hours for one-round transmission of GPT-3 (Tang et al., 2024a)), criticized for being prohibitive in real-world applications (Zhang et al., 2022a; Dai et al., 2023; Chen et al., 2023; Tang et al., 2024a; Liu et al., 2024).

To reduce the communication costs, one-shot Federated Learning (OFL) has emerged recently by compressing the multi-round communications of vanilla FL into just *one round* (Guha et al., 2019). In OFL, clients perform long-term local training individually based on their private data and upload these well-trained local models to the server for aggregation. In this manner, OFL is believed to be well-suited for the prevalent model market scenarios (Zhang et al., 2022a; Zeng et al., 2024; Liu et al., 2025), where users are willing to trade their models for next-stage knowledge fusion instead of joining into a redundant training process. In the current OFL, clients are supposed to train locally for only one time, gaining one-shot local models, while the server is expected to provide a deliberate aggregation by reforming a global model with one-shot local ones.

Existing OFL aggregation designs, building on optimization-based methods (McMahan et al., 2017; Zeng et al., 2021; Jhunjhunwala et al., 2024; Liu et al., 2024; Dennis et al., 2021; Su et al., 2023), distillation-based methods (Li et al., 2021b; Zhou et al., 2020; Yang et al., 2023), generative methods (Yang et al., 2024b; Heinbaugh et al., 2022), and selective ensemble methods (Zhang et al., 2022a; Dai et al., 2023; Zeng et al., 2024), focus solely on the server side. Unfortunately, there is no free lunch to reduce the cost of communication. The reported performance of OFL methods has a significant gap (over 30%) compared with multi-round FL (Tang et al., 2024a; Zeng et al., 2021; Gao et al., 2022).

This work identifies, for the first time, that existing OFL falls in a garbage (inferior one-shot local models) in and garbage (degraded global model) out pitfall. Vanilla FL

---
[*]Equal contribution  [1]College of Computer Science and Technology, National University of Defense Technology, Changsha, China  [2]National Engineering Research Center for Multimedia Software, School of Computer Science, Wuhan University, Wuhan, China. Correspondence to: Tongqing Zhou <zhoutongqing@nudt.edu.cn>, Zhiping Cai <zpcai@nudt.edu.cn>.

*Proceedings of the 42$^{nd}$ International Conference on Machine Learning*, Vancouver, Canada. PMLR 267, 2025. Copyright 2025 by the author(s).

overcomes such a pitfall by implicitly exchanging local knowledge via multi-round iterations while existing OFL lacks such a measure by reactively gathering the garbage inputs. We further unravel the root cause of such garbage inputs as two aspects of inconsistency in the face of data heterogeneity (§ 3). (1) Intra-model inconsistency. A one-shot local model is shown to have divergent predictions for samples under the same semantics. (2) Inter-model inconsistency. Different one-shot local models from different clients manifest distinct parameters, causing divergent predictions for even the same sample.

Intuitively, the OFL paradigm would be more effective if provided with one-shot but consistent local models. However, dealing with inconsistencies in OFL is non-trivial. On one hand, we find that the heterogeneity of different categories of one client's samples leads to intra-model inconsistency. *How can we construct a model capable of capturing invariant features and achieving stable predictions under such heterogeneous conditions* (**Challenge #1**)? On the other hand, we observe that better local performance is always accompanied by larger parameter discrepancies. A more tricky challenge is *how can we effectively leverage models with parameter discrepancies in a one-shot manner* (**Challenge #2**)?

In view of these challenges, this work presents a novel one-shot Federated Learning framework, named FAFI. For **Challenge #1**, we design Self-Alignment Local Training (SALT), a dual-step training strategy, which leverages contrastive learning to capture invariant feature representations and employs a learnable category-wise prototype to address the prediction inconsistency by establishing semantically aligned decision boundary. Working in an unsupervised manner, SALT could improve the generalization of the one-shot local training with a tuned feature extractor and category-wise templates (i.e., prototypes).

For **Challenge #2**, instead of directly utilizing divergent parameters, FAFI performs informativeness-aware feature fusion based on local-uploaded extractors and aggregate prototypes for each category during inference, namely, Informative Feature Fused Inference (IFFI) in the server. Specifically, to reduce the impact of local extractors with uncertain understanding of a sample, the sample's representation from each extractor is compared with a noise representation of the same extractor. Those with noise-proximal features are assigned with attenuated attention during fusion. Subsequently, the inference is made by finding the nearest aggregated prototype for the fused feature of that sample.

Our main contributions are summarized as follows:

• We empirically and theoretically reveal model inconsistencies within and across local models, which leads to poor performance for existing model merging designs in OFL.

• We present the novel one-shot Federated Learning design

of client-side Self-Alignment Local Training and server-side Informative Feature Fused Inference (FAFI), mitigating inconsistency in OFL for enhanced inference performance without requiring additional sample or local model exchanging.

• We conduct extensive evaluations on real datasets with various levels of data heterogeneity, accompanied by a set of ablative studies. Experimental results demonstrate the performance superiority of the proposal (10.86% accuracy improvement over 11 baselines on 3 datasets) and the indispensability of each module.

## 2. One-shot Federated Learning

One-shot Federated Learning (OFL) is a variant of FL that requires only one round of interaction between the clients and server to reduce the heavy communication cost. Existing OFL methods focus on designing aggregation mechanisms on the server side, which can be categorized into three categories.

**(1) Optimization-based** methods focus on reconstructing a better global model with the parameters of local models. Traditional aggregation methods such as FedAvg (McMahan et al., 2017), Median (Yin et al., 2018), Krum (Blanchard et al., 2017), and FedCav (Zeng et al., 2021; 2025) can be directly applied in OFL, but achieve low performance. MA-Echo (Su et al., 2023) tries to get the Pareto optimum of the local clients via exploring common harmonized optima. FedFisher (Jhunjhunwala et al., 2024) and FedLPA (Liu et al., 2024) require additional Fisher information matrices for reaggregation. However, the nonlinear structure of DNNs makes it difficult to obtain a comparable global model through parameter optimization (Tang et al., 2024a). Besides, directly analyzing the model parameters requires all local models to have the same architecture, whose setting is impractical in real-world heterogeneous scenarios.

**(2) Distillation-based** methods try to introduce knowledge distillation (KD) to transfer the massive local knowledge into one global model. The local models (Guha et al., 2019; Li et al., 2021b; Diao et al., 2022) or locally distilled data (Zhou et al., 2020) are viewed as teachers and the newly constructed global model is the student. (Guha et al., 2019) firstly uses the ensemble prediction of local models as the teachers' output. FedKT (Li et al., 2021b) designs a two-tier PATE structure relying on public data to improve the ensemble of local models. To alleviate the label skews, FedOV (Diao et al., 2022) adopts open-set voting in OFL to enhance the generalization. k-FED (Dennis et al., 2021) runs a variant of Lloyd's method for k-means clustering and obtains an aggregated model through one round iteration of exchanging local cluster means. Some dataset distillation-based one-shot methods, such as FedD3 (Song et al., 2023) and FedMD (Li & Wang, 2019), transmit the

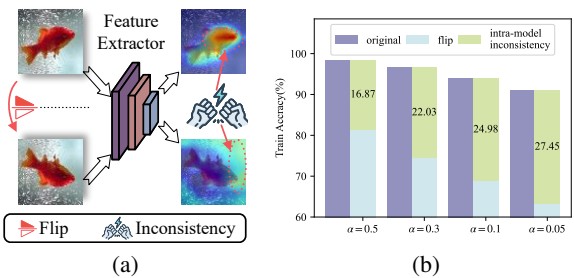

*Figure 1.* **Empirical demonstration of intra-model inconsistency.** (a) The inconsistent features extracted by the one-shot local model and (b) the inconsistent predictions of the one-shot local model on the original and flipped samples.

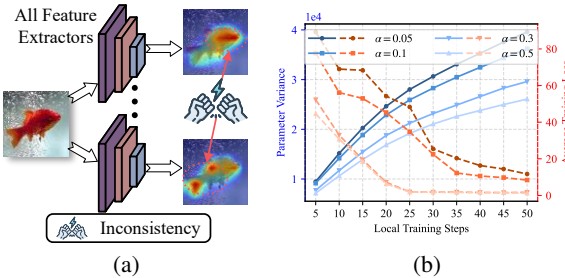

*Figure 2.* **Empirical demonstration of inter-model inconsistency.** (a) The inconsistent features across different one-shot local models. (b) The inconsistent parameters (measured by parameter variance) and averaged training loss under different data heterogeneity. The solid blue lines represent the parameter variance, and the dotted red lines represent the average training loss.

locally distilled dataset rather than models to the server in a one-shot manner. However, these methods require auxiliary public data or pre-trained models which may be impractical in privacy-sensitive scenarios, such as biomedical domains.

**(3) Generative** methods use the generative models to synthesize proxy samples for centralized training on the server side. FedCAVE (Heinbaugh et al., 2022) modifies the local learning task into training a conditional variation auto-encoder (CAVE) and uses KD to compress the ensemble into a powerful decoder. The decoder can be used to generate training samples for the global model. FedCADO (Yang et al., 2023) adopts the popular diffusion models to get the synthetic data. FedDEO (Yang et al., 2024b) trains local descriptions that serve as the medium for conditional generation with diffusion models. However, these generative data samples may leak the privacy of the local clients.

**(4) Selective ensemble-based** methods try to assign proportions for each local model and use them for prediction based on ensemble learning. DENSE (Zhang et al., 2022a) and Ensemble (Wang et al., 2023) equally average the predictions of all local models. Co-Boosting (Dai et al., 2023) designs a learnable weight for each local model and synthesizes data and the ensemble model mutually enhances each other progressively. IntactOFL (Zeng et al., 2024) trains a MoE for dynamic routing. However, all these methods focus on improving performance through server-side design, ignoring that the root cause of low performance in OFL is the inconsistent one-shot local models.

## 3. Motivation

Existing OFL aggregation methods focus on better model merging designs on the server side, which easily falls in a garbage in garbage out pitfall. In this part, we empirically and theoretically unravel the root cause of such 'garbage input' as two aspects of inconsistency (intra-model and inter-model levels) in the face of data heterogeneity.

### 3.1. Intra-model Inconsistency

In this part, we first investigate the intra-model inconsistency in existing one-shot local models in heterogeneous data, which is a common issue in FL, that the categories of one client's training samples are totally different (Luo et al., 2021; Li et al., 2023).

**Empirically Demonstration.** As shown in Figure 1a and Figure 1b, the one-shot local model exhibits significant inconsistencies in their extracted feature (visualized by Grad-CAM (Zhou et al., 2016)) and prediction for identical samples when subjected to simple augmented transformations, such as flipping. Besides, with the increased non-IID degree (lower Dirichlet Distribution parameter $alpha$), the intra-model inconsistency (measured by the performance gap between original and flipped samples) becomes more significant. We attribute this to the heterogeneity of local data, which hinders the acquisition of a one-shot local model with good generalizability.

**Theoretical Analysis.** We represent the $m^{th}$ client local model $w_m = \theta_m \cdot \Phi_m$, where $\Phi_m$ is the classifier and $\theta_m$ is the feature extractor. Existing OFL methods primarily train local models in a supervised manner, using the cross-entropy loss as the supervised loss. Considering a classification task with $C$ categories, the loss can be represented as:

$$\mathcal{L}_i^{sup}(w_m) = \mathbb{E}_{(x,y)\in\mathcal{D}_m}\left[\ell_m^{sup}(\Phi_m, \theta_m; (x,y))\right]$$
$$= -\frac{1}{n_m}\sum_{j=1,y_j=c}^{n_m}\log\frac{\exp(\Phi_{m,c}^T\theta_{m,c}^Tx_j)}{\sum_{k=1}^C\exp(\Phi_{m,k}^T\theta_{m,k}^Tx_j)}, \quad (1)$$

where $n_m$ is the number of samples in the $m$-th client. The one-shot local model should have consistent predictions for samples with the same semantics. Thus, we define the intra-model inconsistency as the performance discrepancy of the model on the original samples $(x, y)$ and augmented samples $(A(x), y_a)$, where $A$ is the data augmentation function,

such as RotateX/Y and Flip. We assume that the augmented samples have significant differences from the original samples without losing semantics (Cao et al., 2024), that is, $\|A(x) - x\| > 0, y = y_a$.

We use the $\Delta_{intra} = |\mathcal{L}(x, y) - \mathcal{L}(x', y)|$, which is the performance discrepancies between any two samples $x, x'$ with the same label $y$. We have the following analysis.

**Theorem 3.1.** *(Intra-model inconsistency. See proof in Appendix A). The intra-model inconsistency of the one-shot local model on the original samples $(x, y)$ and augmented samples $(A(x), y)$ can be represented as:*

$$\|\Delta_{intra}\|^2 \geq \| (p \cdot \nabla g_a \cdot \nabla A)^T (x - A(x))\|^2 > 0, \quad (2)$$

*where $p = \sum_{c=1}^{C}(z_c - y_c)$, $z$ is the prediction of $w_i$ activated by softmax function with the augmented samples $A(x)$, $\nabla g_a$ is the gradient of the local model $w_i$, and $\nabla A$ is the gradient of the data augmentation function.*

Theorem 3.1 indicates that the intra-model inconsistency is inevitable in the one-shot local model when trained through existing OFL paradigms on heterogeneous data. Specifically, it is mainly caused by three factors. (1) The performance on augmented samples, which is represented by $p\nabla g_a$. (2) The transformations property of the data augmentation function, which is represented by $\nabla A$. (3) The discrepancy between the original samples and augmented samples, which is represented by $(x - A(x))$. However, all these factors are larger than zero, leading to the existence of intra-model inconsistency.

### 3.2. Inter-model Inconsistency

Notably, inconsistency exists not only within individual one-shot local models but also significantly across multiple models trained on different clients.

**Empirically Demonstration.** We visualize the features extracted by different local models for the same sample in Figure 2a. We observe that the features extracted by different local models are significantly different. Besides, we use parameter variance $\sigma_w = \sqrt{\frac{1}{M} \sum_{i=1}^{M}(w_i - \hat{w})^2}$ to measure the inter-model inconsistency of one-shot local models. As demonstrated in Figure 2b, we observe that inter-model inconsistency increases continuously during the training process, even when the loss is close to convergence. Existing research, such as 'client drift ' (Gao et al., 2022; Karimireddy et al., 2020; Huang et al., 2023a; 2022), discusses the optimization direction inconsistency in multi-round short-term local training scenarios. Differently, our observation focuses on one-shot long-term local training, where better performance is always accompanied by larger inter-model inconsistency.

**Theoretical Analysis.** Let $z_c^j = \frac{exp(w^T x_c)}{\sum_{k=1}^{C} exp(w_k^T x_k)}$, where

$\mathcal{F}_c$ is the feature of the $c$-th class and $w = \theta \cdot \Phi$ is the one-shot local model. We denote $\overline{x}_c$ and $\overline{z}$ as the average input samples and prediction of all local models, respectively.

**Theorem 3.2.** *(Inter-model inconsistency. See proof in Appendix B). For any two client $u$ and $v$ with the same quantity of samples $n_u = n_v$, the one-step model deviation between the two clients $\Delta_{inter} = \nabla w_u - \nabla w_v$ can be represented as:*

$$\begin{aligned}
&\|\Delta_{inter}\|^2 \\
&= \|\frac{\eta}{n_u}[(n_{u,c}(1 - \overline{z}_{u,c})\overline{x}_{u,c} - n_{v,c}(1 - \overline{z}_{v,c})\overline{x}_{v,c}) \\
&\quad - (\sum_{c' \in [C_u]\backslash c} n_{u,c'}\overline{z}_{u,c'}\overline{x}_{u,c'} - \sum_{c' \in [C_v]\backslash c} n_{v,c'}\overline{z}_{v,c'}\overline{x}_{v,c'})]\|^2 \\
&> 0,
\end{aligned}$$
$$(3)$$

*where $\eta$ is the learning rate, $n_{u,c}$ and $n_{v,c}$ is the sample quantity of $c$-th class, $c'$ is the negative classes except $c$.*

Theorem 3.2 unravels that for any two clients, each local training step would cause the inconsistency of one-shot local models. With more local training steps $E$, the inter-model inconsistency becomes more significant, which can also be verified in Figure 2b.

In this section, we empirically and theoretically demonstrate the negative effect on one-shot local models at both intra-model levels and inter-model levels. We show that the current OFL local training strategies inevitably cause 'garbage' one-shot local models, making it challenging for server-side aggregation.

## 4. Method

### 4.1. Overview

Motivated by the analysis in § 3, we propose a novel OFL framework, namely FAFI. It consists of two components: self-alignment local training and informative feature fused inference. (1) On the client side, we train the feature extractor to enable the model to learn invariant features that can be generalized to diverse augmented samples. We also design category-wise prototype learning for distinctive prototypes, replacing the original classifier, thereby mitigating the negative impact on the prediction. (2) On the server side, we aggregate the prototypes from all clients into a global prototype. During the inference stage, we informatively fuse the features extracted by the local models to alleviate the inter-model inconsistency. The overview of FAFI is illustrated in Figure 3.

### 4.2. Self-alignment Local Learning

**Motivation.** According to Theorem 3.1, we note that the key factor leading to intra-model inconsistency is the model's

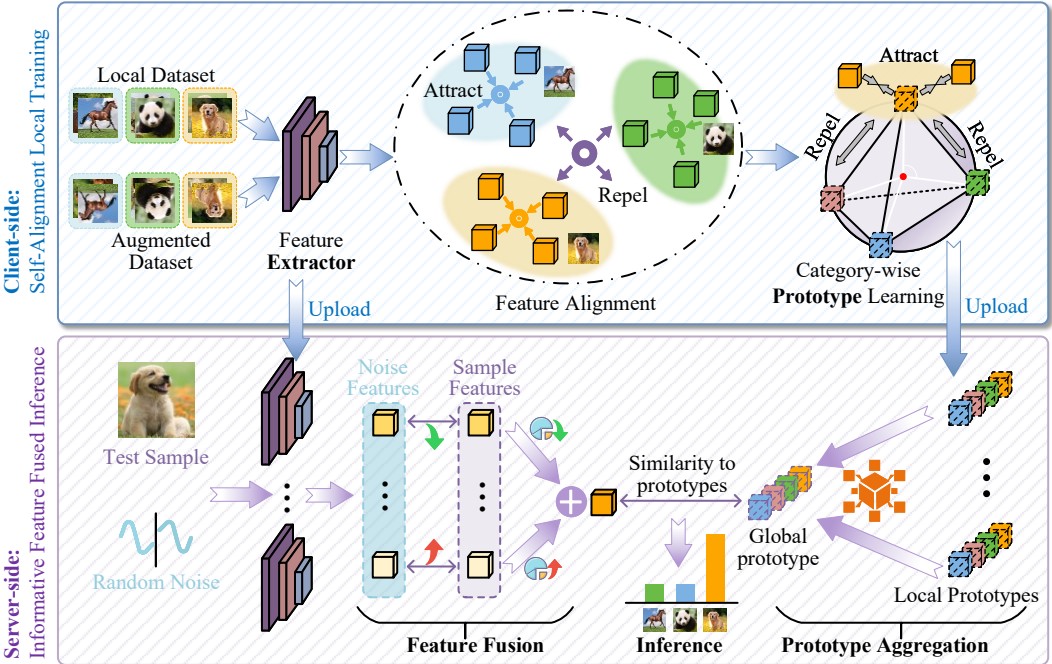

*Figure 3.* An overview of the proposed FAFI. All clients perform local training and upload the trained models to the server once. (1) Through self-alignment local training, clients could upload generalizable feature extractors and category-wise prototypes. (2) The server aggregates all local prototypes into a global prototype, while in the inference stage, the server informative fuses the features extracted from the local models. It is best viewed in color.

inability to handle augmented samples with the same semantics, i.e., when $\Delta g_a \gg 0$. Unfortunately, the current supervised learning paradigm can only learn fixed semantics based on labels and original input, lacking generalization to diverse augmented samples. If the model can perform well on any augmented samples, i.e., $\Delta g_a = 0$, then the intra-model inconsistency will be alleviated. Intuitively, it seems that directly adopting data augmentation could resolve this problem. However, constrained by the supervised training paradigm, as long as training with labels, $p \neq 0$, the intra-model inconsistency will inevitably occur. Besides, existing works have shown that solely adopting data augmentation can lead to the problem of *complete collapse* (Grill et al., 2020). To this end, we introduce self-alignment learning to learn invariant features and an unbiased classifier that can generalize to diverse augmented samples.

**Feature Alignment.** To enable the model to learn more generalized features and reduce the impact of biased data, we focus on learning invariant features related to themselves rather than features aligned with labels. We introduce a contrastive learning approach. Specifically, consider the $m^{th}$ clients with its local dataset $\mathcal{D}_m = \{x_i, y_i\}_{i=1}^{n_m}$ and the augmented dataset $\mathcal{D}_m^a = \{A(x_i), y_i\}_{i=1}^{n_m}$, where $A$ denotes the augmentation operator. Through self-supervised learning, we aim to learn invariant features $\mathcal{F}_{x_i} = f(\theta_m; x_i)$ and $\mathcal{F}_{A(x_i)} = f(\theta_m; A(x_i))$ that can generalize to diverse

augmented samples. With similarity function $sim$, the self-supervised learning loss, whose objective is to minimize the discrepancy with the same semantics and different the representation to all other semantics, can be formulated as follows:

$$\mathcal{L}_{\text{ssl}} = -\frac{1}{n_m} \sum_{i=1}^{n_m} \log \frac{s(\mathcal{F}_{x_i}, \mathcal{F}_{x_i^+})}{\sum_{j \in N_g(y_i)} s(\mathcal{F}_{x_i}, \mathcal{F}_{x_j})}, \quad (4)$$

where $s(\mathcal{F}_{x_i}, \mathcal{F}_{x_i^+}) = exp(cos(\mathcal{F}_{x_i}, \mathcal{F}_{x_i^+})/\tau)$, $\tau$ denotes the temperature parameter, $x_i^+$ represents the set of samples that have the same label with $x_i$, $N_g(y_i)$ denotes the set of sample indexes that are different from $y_i$, and the $cos$ function is cosine similarity.

**Category-wise Prototype Learning.** While contrastive learning can achieve invariant features, the classifier often displays biased behaviors and is sensitive to data heterogeneity, resulting in inconsistent predictions. Inspired by the success of prototype learning in various heterogeneous data scenarios, the learnable contrastive prototypes replace the original classifier, thereby mitigating the negative impact of data heterogeneity on the classification (Zhang et al., 2024; Wan et al., 2024a). Our goal is to obtain a set of representative and highly discriminative prototypes. (1) Closely align with the features to retain semantic information. (2) maintain category-wise distinctions between each prototype. Specifically, we define each client maintains a set

of prototypes $\mathcal{P}_m = \{p_{m,1}, p_{m,2}, \ldots, p_{m,C}\}$, where $C$ is the number of classes. During local training, the prototypes are updated to minimize the contrastive loss between the features extracted by the local model and the prototypes. The contrastive loss for the $m^{th}$ client can be formulated as:

$$\mathcal{L}_{\text{proto}} = -\frac{1}{n_m} \sum_{i=1}^{n_m} \log \frac{\exp(\mathcal{F}_{x_i}^T p_{m,y_i}/\tau)}{\sum_{j \in [C \setminus y_i]} \exp(\mathcal{F}_{x_i}^T p_{m,j}/\tau)}, \quad (5)$$

where $\mathcal{F}_{x_i}$ is the feature extracted by the local model for sample $x_i$, $p_{u,y_i}$ is the prototype corresponding to the ground truth class $y_i$, and $\tau$ is the temperature parameter.

**Overall Objective.** Thus, the overall local training objective can be:

$$\mathcal{L}_{\text{local}} = \mathcal{L}_{\text{ssl}} + \mathcal{L}_{\text{proto}}. \quad (6)$$

Notably, the objective of self-supervised local learning loss is consistent with alleviating inter-model inconsistency. And the two losses in it are complementary to each other. For learnable category-wise prototypes $\mathcal{P}_u$, the prediction of the original sample and augmented sample can be formulated as $\mathcal{F}_{x_i}^T \mathcal{P}_u$ and $\mathcal{F}_{A(x_i)}^T \mathcal{P}_u$, respectively. Using the cosine similarity as the distance metrics, the prediction discrepancy, i.e., inter-model inconsistency, can be formulated as:

$$\Delta_{intra} = -cos( \underbrace{\mathcal{F}_{x_i}^T \mathcal{P}_m}_{\mathcal{L}_{\text{proto}} \propto}, \underbrace{\mathcal{F}_{A(x_i)}^T \mathcal{P}_m}_{\mathcal{L}_{\text{proto}} \propto} )$$
$$= - \underbrace{cos(\mathcal{F}_{x_i}, \mathcal{F}_{A(x_i)})}_{\mathcal{L}_{\text{ssl}} \propto} \mathcal{P}_m, \quad (7)$$

we can observe that the optimization objective of $\mathcal{L}_{\text{proto}}$ in Eq.(5) is to learn a discriminative representation, ensuring that the samples with the same semantics have similar predictions, which aligns with the goal of reducing intra-model inconsistency (first line in Eq.(7)). Simultaneously, the objective of $\mathcal{L}_{\text{ssl}}$ in Eq.(4) is to enable the local model to learn informative and consistent features (second line in Eq.(7)), which also help to reduce intra-model inconsistency.

## 4.3. Informative Feature Fused Inference

**Motivation.** The aforementioned theoretical analysis Theorem 3.2 and recent studies indicate that inter-model inconsistency caused by data heterogeneity is inevitable. The root cause of the low performance due to inter-model inconsistency lies in the inability to reconstruct a well-performing global model through parameter-level aggregation from significantly different local models. Inspired by the Mixture of Experts (MoE) (Zeng et al., 2024; Zhu et al., 2024b), an architecture that enhances the inference capability by leveraging the outputs of expert models. Instead of aggregating model parameters, we fuse the features extracted by inconsistent local models to integrate semantics, thereby

mitigating the negative impact of data heterogeneity and enhancing the inference capability. Additionally, we also note that the features extracted by different models exhibit discrepancies during the fusion process. To address this, we design an attention-based feature fusion mechanism.

**Feature Fusion.** Specifically, let $\mathcal{F}_m$ be the feature extracted by the $u$-th client's local model. To fuse the features from different clients, we design a mechanism to informatively aggregate the features. The features with less information, which are similar to noise should be down-weighted, while the features with more information should be up-weighted. Thus, we define the rescaling factor $\alpha_u$ as:

$$\alpha_m = 1 - cos(\mathcal{F}_m, \mathcal{F}_m^{\mathcal{N}(\mu,\sigma)}), \quad (8)$$

where $sim(\cdot, \cdot)$ is the cosine similarity, and $\mathcal{F}_u^{\mathcal{N}(\mu,\sigma)}$ is a feature extracted by a Gaussian distribution with mean $\mu$ and standard deviation $\sigma$. We use the standard Gaussian distribution $\mathcal{N}(0,1)$ in our implementation.

Next, we aggregate the features from different clients using the weighted average:

$$\mathcal{F}_{\text{fused}} = \sum_{m}^{M} \frac{\alpha_m}{\sum_{v}^{M} \alpha_v} \mathcal{F}_m, \quad (9)$$

where $M$ is the number of clients.

**Inference with Global Prototype.** After obtaining the fused feature $\mathcal{F}_{\text{fused}}$, we use the similarity between the fused feature and the discriminative global prototype for prediction. Specifically, let $\mathcal{P}_g = \frac{1}{M} \sum_{m=1}^{M} \mathcal{P}_m$ be the global prototype aggregated from the clients' learnable prototypes. The prediction $\hat{y}$ can be formulated as:

$$\hat{y} = \arg\max_{c \in [C]} cos(\mathcal{F}_{\text{fused}}, \mathcal{P}_{g,c}), \quad (10)$$

where $cos(\cdot, \cdot)$ is the cosine similarity. This approach leverages the fused features and the discriminative power of the global prototype to make accurate predictions. The detail process is presented in Appendix C Algorithm 1.

## 4.4. Discussion

**Privacy Security.** Transmitting prototypes from clients to the server instead of classifier is a common practice in FL (Tan et al., 2022; Mu et al., 2023) that does not compromise privacy security, as prototypes are statistical-level information of category that does not contain the privacy of individual samples (Huang et al., 2023b; Wan et al., 2024b). Besides, category-wise prototype learning aims to capture the common abstract features of the same category, thereby reducing the exposure risk for individual samples.

**Comparison with Analogous Methods.**

• **Prototypes in FL.** Existing prototype-based methods, such as FedProto (Tan et al., 2022) and FedTGP (Zhang et al., 2024), rely on **multi-round interactions** to obtain a representative global prototype, which limits their applicability in one-shot FL scenarios. In contrast, FAFI can achieve a semantically aligned global prototype with just one-shot aggregation.

• **Model Merging in LLMs.** Some studies consider that the existing aggregation process in OFL is akin to model merging and try to improve the performance by either using weighted-based model merging (Tao et al., 2024; Yang et al., 2024a), subspace-based merging (Zhu et al., 2024a; Yadav et al., 2024), or routing-based merging (Zeng et al., 2024). However, (1) all these methods still suffer from the 'garbage in, garbage out' pitfall, as they only focus on server-side merging and overlook the negative effect caused by inconsistent pre-trained models; (2) All these methods lack privacy-preserving properties, as they require source data or additional information that is highly relevant for post-calibration after merging. In contrast, FAFI does not require the source data or any other auxiliary information.

**Limitations.** Our framework exhibits limitations in domain shift and multi-task scenarios where static class representations fail to adapt to conflicting feature distributions or divergent task objectives. This limitation is shared by other prototype-based methods. (Tan et al., 2022; Huang et al., 2023b; Zhang et al., 2024; Mu et al., 2023). One possible solution for this limitation is to cluster and ensemble feature representations across various domains or tasks, allowing unbiased representations to encapsulate multi-domain/task knowledge (Huang et al., 2023b).

# 5. Experiments

## 5.1. Experimental Setup

**Datasets.** Adhere to the previous work (Tan et al., 2022; Huang et al., 2023b; Zeng et al., 2024; Zhang et al., 2022a; Tang et al., 2024a), we evaluate the efficacy on three widely used benchmarks:

• **CIFAR-10** contsins 50k, 10k images for training and testing. Images are in size $32 \times 32$ with 10 classes.
• **CIFAR-100** have the same format and size as CIFAR-10, but with 100 classes.
• **Tiny-Imagenet** contains 100k, 10k images for training and testing. Images are in size $64 \times 64$ with 200 classes.

**Data Heterogeneity.** Considering the heterogeneous environment, we partitioned the dataset through a widely-used non-IID partition method, namely Dirichlet Sampling, in which the coefficient $\alpha$ refers to the non-IID degree. A small $\alpha$ represents a biased distribution. Following the setting in (Zeng et al., 2024; Zhang et al., 2022a; Zeng et al., 2022), we set $\alpha \in \{0.05, 0.1, 0.3, 0.5\}$, respectively. More details are shown in Appendix E.

**Baselines.** We compare with several OFL methods, categorized into three types:

• **Optimization-based**: one-shot FedAvg (O-FedAvg) (McMahan et al., 2017; Guha et al., 2019) [arXiv'19], MA-Echo (Su et al., 2023) [NN'23], and FedFisher (Jhunjhunwala et al., 2024) [AISTATS'24].
• **Distillation-based**: DENSE (Zhang et al., 2022a) [NeurIPS'22], FedDF (Lin et al., 2020) [NeurIPS'20], F-ADI (Yin et al., 2020) [CVPR'20], and F-DAFL (Chen et al., 2019) [ICCV'19].
• **Selective ensemble learning-based**: directly Ensemble, Co-Boosting (Dai et al., 2023) [ICLR'23], IntactOFL (Zeng et al., 2024) [MM'24].

Besides, we also consider some iterative methods, i.e., FuseFL (Tang et al., 2024a) [NeurIPS'24], which improves the performance through iteratively sharing intermediate features. To ensure fair comparisons, we neglect some methods that require additional information, such as FedKT (Li et al., 2021b), FedOV (Diao et al., 2022), and FedGen (Zhu et al., 2021).

## 5.2. Effectiveness

Table 1 illustrates the effectiveness of our proposed FAFI compared with popular OFL methods in non-IID settings. (1) It clearly depicts that our method achieves a significant performance improvement over the baselines on all datasets and all settings (10.86% averaged). (2) Notably, in some extreme cases such as $\alpha = \{0.05, 0.1\}$ on Tiny-Imagenet, our method exhibits a more significant performance advantage (17.71% averaged) over the baseline algorithms. (3) Besides, the FuseFL which iteratively shares intermediate features between clients can achieve the second-best performance in some settings. We attribute this to the fact that the FuseFL mitigates the feature inconsistencies by sharing intermediate features, leaving intra-model inconsistency unsolved. **In summary, the FAFI is effective in various data heterogeneous scenarios and achieves competitive performance over all baselines.**

## 5.3. Scalability

We assess the scalability of FAFI by varying the number of clients $m$. As presented in Table 2, FAFI consistently achieves the best performance across different client scales. As suggested in (Lian et al., 2017), the server can become a major bottleneck while the number of clients increases. We also reach a similar conclusion, with more clients $m$, the performance decreases, which is consistent with (Zhang et al., 2022a; Dai et al., 2023; Lian et al., 2017; Zeng et al., 2024). **In summary, the FAFI is scalable across diverse distributed networks of varying sizes.**

*Table 1.* **Comparison with the state-of-the-art OFL methods**: in CIFAR-10, CIFAR-100, and Tiny-ImageNe scenarios with skew ratio $\alpha \in \{0.05, 0.1, 0.3, 0.5\}$. Underline/**bold** fonts highlight the best baseline/the proposed FAFI. $\Delta$ represents the performance improvement compared with the best baseline. We report the 5 trials' results in the form of mean±variance. See details in § 5.2.

| Methods | CIFAR-10 | | | | CIFAR-100 | | | | Tiny-ImageNet | | | |
|---|---|---|---|---|---|---|---|---|---|---|---|---|
| | $\alpha = 0.05$ | $\alpha = 0.1$ | $\alpha = 0.3$ | $\alpha = 0.5$ | $\alpha = 0.05$ | $\alpha = 0.1$ | $\alpha = 0.3$ | $\alpha = 0.5$ | $\alpha = 0.05$ | $\alpha = 0.1$ | $\alpha = 0.3$ | $\alpha = 0.5$ |
| MA-Echo | $36.77_{\pm0.91}$ | $51.23_{\pm0.28}$ | $60.14_{\pm0.21}$ | $64.21_{\pm0.23}$ | $19.54_{\pm0.45}$ | $29.11_{\pm0.26}$ | $37.77_{\pm0.24}$ | $41.94_{\pm0.21}$ | $15.46_{\pm0.66}$ | $22.23_{\pm0.56}$ | $23.46_{\pm0.19}$ | $28.21_{\pm0.42}$ |
| O-FedAvg | $12.13_{\pm2.11}$ | $17.43_{\pm0.51}$ | $28.07_{\pm0.89}$ | $35.42_{\pm0.67}$ | $4.77_{\pm0.21}$ | $6.45_{\pm0.71}$ | $10.67_{\pm0.31}$ | $12.13_{\pm0.05}$ | $5.67_{\pm0.45}$ | $8.31_{\pm0.21}$ | $13.61_{\pm0.10}$ | $13.71_{\pm0.16}$ |
| FedFisher | $40.03_{\pm1.11}$ | $47.01_{\pm1.81}$ | $49.33_{\pm1.52}$ | $50.34_{\pm1.32}$ | $16.56_{\pm2.67}$ | $18.98_{\pm2.09}$ | $27.24_{\pm1.92}$ | $31.44_{\pm1.87}$ | $15.65_{\pm1.54}$ | $17.89_{\pm1.46}$ | $19.54_{\pm1.31}$ | $20.77_{\pm1.15}$ |
| FedDF | $35.53_{\pm0.67}$ | $41.58_{\pm0.80}$ | $44.78_{\pm0.60}$ | $54.58_{\pm0.73}$ | $15.07_{\pm0.74}$ | $27.17_{\pm0.55}$ | $31.23_{\pm0.79}$ | $35.39_{\pm0.47}$ | $11.45_{\pm0.40}$ | $16.32_{\pm0.33}$ | $17.79_{\pm0.57}$ | $27.55_{\pm0.66}$ |
| F-ADI | $35.93_{\pm1.56}$ | $48.35_{\pm1.23}$ | $52.66_{\pm1.44}$ | $58.78_{\pm1.67}$ | $14.65_{\pm0.98}$ | $28.13_{\pm1.24}$ | $33.18_{\pm0.67}$ | $39.44_{\pm1.11}$ | $13.92_{\pm1.99}$ | $19.00_{\pm1.78}$ | $26.01_{\pm1.44}$ | $29.98_{\pm1.34}$ |
| F-DAFL | $38.32_{\pm1.40}$ | $46.34_{\pm1.12}$ | $54.03_{\pm1.71}$ | $59.09_{\pm2.23}$ | $16.31_{\pm0.33}$ | $26.80_{\pm1.33}$ | $34.89_{\pm1.45}$ | $37.88_{\pm1.34}$ | $15.12_{\pm1.34}$ | $19.01_{\pm1.11}$ | $23.78_{\pm1.23}$ | $27.98_{\pm1.10}$ |
| DENSE | $38.37_{\pm1.08}$ | $50.26_{\pm0.24}$ | $59.76_{\pm0.45}$ | $62.19_{\pm0.12}$ | $18.37_{\pm2.43}$ | $32.03_{\pm0.44}$ | $37.33_{\pm0.48}$ | $38.84_{\pm0.39}$ | $18.77_{\pm0.67}$ | $22.25_{\pm0.33}$ | $28.14_{\pm0.34}$ | $32.34_{\pm0.32}$ |
| Ensemble | $41.36_{\pm0.67}$ | $45.43_{\pm0.32}$ | $62.18_{\pm0.34}$ | $61.61_{\pm0.23}$ | $20.46_{\pm0.62}$ | $26.23_{\pm0.65}$ | $38.01_{\pm0.67}$ | $41.61_{\pm0.77}$ | $13.28_{\pm0.67}$ | $15.38_{\pm0.23}$ | $17.53_{\pm0.31}$ | $28.50_{\pm0.46}$ |
| Co-Boosting | $39.20_{\pm0.81}$ | $58.49_{\pm1.24}$ | $67.21_{\pm1.76}$ | $70.24_{\pm2.34}$ | $20.19_{\pm1.44}$ | $27.59_{\pm1.35}$ | $39.30_{\pm1.30}$ | $42.67_{\pm1.40}$ | $19.00_{\pm1.45}$ | $21.90_{\pm1.20}$ | $29.24_{\pm1.32}$ | $30.78_{\pm2.01}$ |
| FuseFL | $54.42_{\pm0.41}$ | $73.79_{\pm0.34}$ | $84.58_{\pm0.91}$ | $84.34_{\pm0.88}$ | $29.12_{\pm0.23}$ | $36.86_{\pm0.38}$ | $45.12_{\pm0.51}$ | $49.30_{\pm0.32}$ | $22.15_{\pm2.11}$ | $29.28_{\pm2.04}$ | $33.04_{\pm1.79}$ | $34.34_{\pm1.81}$ |
| IntactOFL | $48.22_{\pm0.43}$ | $61.13_{\pm0.63}$ | $70.21_{\pm0.60}$ | $79.93_{\pm0.23}$ | $27.99_{\pm0.67}$ | $39.15_{\pm0.46}$ | $41.86_{\pm0.60}$ | $46.78_{\pm0.78}$ | $20.45_{\pm0.34}$ | $28.43_{\pm0.17}$ | $30.15_{\pm0.12}$ | $35.09_{\pm0.14}$ |
| Ours | $\mathbf{71.84}_{\pm1.53}$ | $\mathbf{77.83}_{\pm1.32}$ | $\mathbf{84.76}_{\pm0.46}$ | $\mathbf{88.74}_{\pm0.11}$ | $\mathbf{31.02}_{\pm1.17}$ | $\mathbf{45.48}_{\pm1.01}$ | $\mathbf{56.65}_{\pm0.91}$ | $\mathbf{61.07}_{\pm0.55}$ | $\mathbf{36.96}_{\pm0.92}$ | $\mathbf{43.62}_{\pm0.77}$ | $\mathbf{53.32}_{\pm0.50}$ | $\mathbf{56.48}_{\pm0.32}$ |
| $\Delta$ | ↑ 17.42 | ↑ 6.04 | ↑ 0.18 | ↑ 4.40 | ↑ 1.90 | ↑ 6.33 | ↑ 11.53 | ↑ 11.77 | ↑ 14.81 | ↑ 14.34 | ↑ 20.28 | ↑ 21.39 |

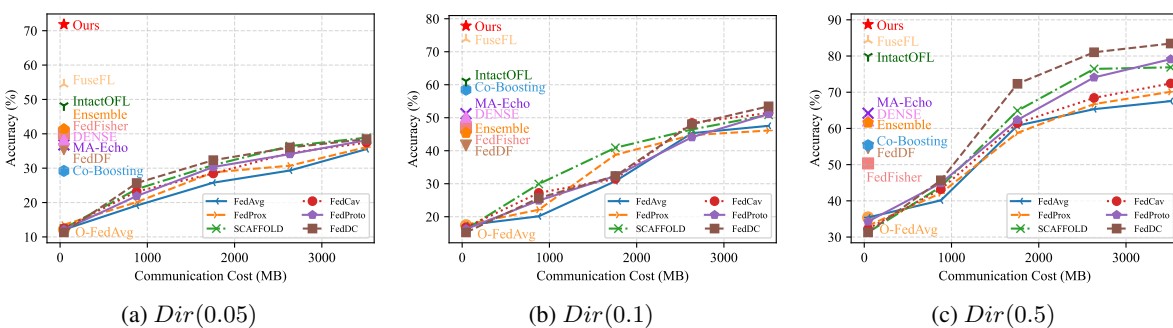

|  (a) $Dir(0.05)$ | (b) $Dir(0.1)$ | (c) $Dir(0.5)$ |
|---|---|---|

*Figure 4.* **Test Accuracy v.s. Communication Cost** on CIFAR-10. FAFI is more efficient than all other baselines, while achieving higher performance without large communication overhead. See details in § 5.4.

*Table 2.* **Scalability under different number of clients**, $m = \{5, 10, 25, 50, 100\}$ on CIFAR-10 with $\alpha = 0.5$. See in § 5.3.

| Methods | Client scales $m$ | | | | |
|---|---|---|---|---|---|
| | 5 | 10 | 25 | 50 | 100 |
| MA-Echo | 64.21 | 52.64 | 48.36 | 45.35 | 38.54 |
| O-FedAvg | 35.42 | 32.09 | 28.03 | 28.24 | 27.14 |
| FedFisher | 50.34 | 45.67 | 34.66 | 29.09 | 28.89 |
| FedDF | 54.58 | 48.88 | 35.44 | 29.91 | 25.66 |
| F-ADI | 59.34 | 46.33 | 31.83 | 27.66 | 24.89 |
| F-DAFL | 58.59 | 45.45 | 32.88 | 29.98 | 28.91 |
| DENSE | 62.19 | 54.67 | 49.32 | 48.67 | 43.34 |
| Ensemble | 61.61 | 60.44 | 58.44 | 52.51 | 45.72 |
| Co-Boosting | 55.34 | 51.11 | 49.32 | 44.56 | 42.45 |
| FuseFL | 84.34 | 78.28 | 62.12 | 42.18 | 37.11 |
| IntactOFL | 79.93 | 69.11 | 64.32 | 59.45 | 53.21 |
| Ours | 88.74 | 86.96 | 85.25 | 81.32 | 75.37 |

*Table 3.* **Ablation Study on Key Components** of FAFI on CIFAR-10 with skew ratio $\alpha \in \{0.05, 0.1, 0.3, 0.5\}$. See details in § 5.5.

| SALT | IFFI | $\alpha = 0.05$ | $\alpha = 0.1$ | $\alpha = 0.3$ | $\alpha = 0.5$ |
|---|---|---|---|---|---|
| | | 12.13 | 17.43 | 28.07 | 35.42 |
| ✓ | | 53.12 | 55.76 | 58.95 | 61.67 |
| | ✓ | 55.23 | 63.75 | 68.49 | 70.44 |
| ✓ | ✓ | 71.84 | 77.83 | 84.76 | 88.74 |

### 5.4. Efficiency

We compare the accuracy and communication cost of FAFI to existing OFL methods and multi-round FL baselines in Figure 4. We select six representative multi-round FL methods, more details are shown in Appendix E. Note that FAFI achieves higher performance than all other baselines, while incurring a lower communication overhead. Notably, with more communication budget, multi-round FL methods can achieve better performance (Figure 4 only presents the first 80 rounds' results of multi-round FL methods). However, as these methods get convergent, the performance improvement diminishes, and communication costs become prohibitive. In summary, **FAFI achieves a better efficiency compared with existing FL methods (both OFL and multi-round FL), making it more suitable for practical deployment**.

### 5.5. Ablation Study

**Key Components.** For thoroughly analyzing the efficacy of each module, we perform an ablation study to investigate the

*Table 4.* **Ablation Study on** $\mathcal{L}_{ssl}$ **and** $\mathcal{L}_{proto}$ of SALT on CIFAR-10 with $Dir(0.1)$. The default loss function is $\mathcal{L}_{ce}$.

| $\mathcal{L}_{ssl}$ | $\mathcal{L}_{proto}$ | Types | CIFAR-10 | CIFAR-100 | Tiny-ImageNet |
|---|---|---|---|---|---|
| | | Classifier | 17.34 | 6.45 | 8.31 |
| ✓ | | Classifier | 22.34 | 12.45 | 10.44 |
| | | Prototypes | 18.23 | 7.12 | 8.92 |
| ✓ | | Prototypes | 50.12 | 31.89 | 22.34 |
| | ✓ | Prototypes | 52.34 | 33.34 | 23.12 |
| ✓ | ✓ | Prototypes | 77.83 | 45.48 | 43.62 |

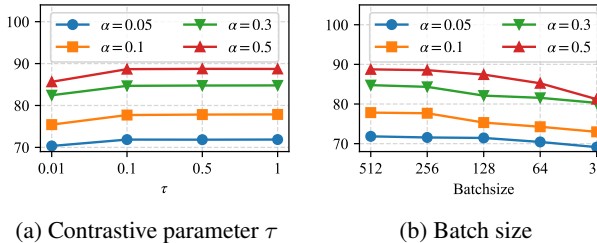

(a) Contrastive parameter $\tau$      (b) Batch size

*Figure 5.* **Analysis on hyper-parameter**. Performance with hyper-parameter $\tau$ and batch size on four data heterogeneity $\alpha \in \{0.05, 0.1, 0.3, 0.5\}$. See details in § 5.5.

effectiveness of Self-alignment Local Training (SALT) and Informative Feature Fused Inference (IFFI). The results in Table 3 show that the two modules contribute significantly to the performance improvement of FAFI. The combination of the two modules achieves the best performance, underscoring the effectiveness of our proposed method.

**Key Loss Functions in SALT.** We investigate two important loss functions ($\mathcal{L}_{ssl}$ and $\mathcal{L}_{proto}$) in SALT. The results in Table 4 present the effectiveness of each loss function. We also find that compared with traditional classifiers, leveraging prototypes helps mitigate model inconsistencies by providing feature-level alignment anchors.

**Hyper-parameter.** We first investigate the $\tau$ and batch size in self-alignment local learning. For $\tau$, we note that its impact on performance when $\tau \in [0.1, 1]$. When $\tau = 0.01$, it would have a certain degree of degradation. For batch size, we note that a larger batch size would benefit the performance while requiring more resources. With the minimum batch size, the performance is still competitive.

## 6. Conclusion

In this paper, we propose a novel OFL framework, namely FAFI, to mitigate the model inconsistency. We first empirically and theoretically present the inconsistencies at both intra-model and inter-model levels in existing OFL methods. To this end, we propose Self-alignment Local Training for semantically aligned feature extractors and distinctive prototypes, and use Informative Feature Fused Inference to fuse the features for inference. Extensive experiments on three datasets show the superiority of our methods.

## Acknowledgements

This work is supported by the National Natural Science Foundation of China (Nos. 62172155, 62472434, 62102425, and 623B2080), the National Key Research and Development Program of China (2022YFF1203001), the Sci-Tech Innovation 2030 Agenda(2023ZD0508600), and the Science and Technology Innovation Program of Hunan Province (Nos. 2022RC3061 and 2023RC3027).

## Impact Statement

This paper presents work whose goal is to advance the field of Machine Learning. There are many potential societal consequences of our work, none of which we feel must be specifically highlighted here.

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

# A. Proof of Theorem 3.1

We first provide a lemma to demonstrate the prediction discrepancy of the one-shot local model on the original sample $(x, y)$ and augmented samples $A(x), y$ as follows.

**Lemma A.1.** *(Prediction discrepancy). For any sample $(x, y)$ and its augmented sample $(A(x), y)$, the prediction discrepancy can be represented as:*

$$\Delta_{intra} \geq (p \cdot \nabla g_a \cdot \nabla A)^T (x - A(x)), \tag{11}$$

*where $p = \sum_{c=1}^{C}(z_c - y_c)$, $z$ is the prediction of $w_i$ activated by softmax function with the augmented samples $A(x)$, $\nabla g_a$ is the gradient of the local model $w_i$, and $\nabla A$ is the gradient of the data augmentation function.*

We provide the proof of Lemma A.1 as follows:

*Proof.* We derive the performance discrepancy of the local model on the original samples $(x, y)$ and augmented samples $(A(x), y)$ as follows.

$$\Delta_{intra} = \mathcal{L}_i^{sup}(w_i; (x, y)) - \mathcal{L}_i^{sup}(w_i; (A(x), y)). \tag{12}$$

We use the Taylor expansion to approximate the loss function on the original samples $(x, y)$ as follows:

$$\mathcal{L}_i^{sup}(w_i; (x, y)) \approx \mathcal{L}_i^{sup}(w_i; (A(x), y)) + \nabla \mathcal{L}_i^{sup}(w_i; (A(x), y))^T (x - A(x)). \tag{13}$$

We can further simplify Equ. 12 as follows:

$$\Delta_{intra} \geq \nabla \mathcal{L}_i^{sup}(w_i; (A(x), y))^T (x - A(x)). \tag{14}$$

If we adopt cross-entropy loss as the supervised loss $\mathcal{L}(z, y) = -\sum_{c=1}^{C} y_k \log z_k$, where $z$ is the softmax prediction, $y$ is the ground truth label, we have:

$$
\begin{aligned}
\Delta_{intra} &\geq \left( \frac{\partial \mathcal{L}}{\partial z} \frac{\partial z}{\partial A} \frac{\partial A}{\partial x} \right)^T (x - A(x)) \\
&\geq \left( \sum_{c=1}^{C}(z_c - y_c) \frac{\partial z}{\partial A} \nabla A \right)^T (x - A(x)).
\end{aligned}
\tag{15}
$$

Note that $\frac{\partial z}{\partial A}$ is the gradient $\nabla g$ of local model $w_i$ with respect to the augmented data $A(x)$, and $\nabla A = \frac{\partial A}{\partial x}$ is the transformations property of the data augmentation function $A$. Since $z_c = \frac{exp(\Phi_{i,c}^T \theta_{i,c}^T A(x_j))}{\sum_{k=1}^{C} exp(\Phi_{i,k}^T \theta_{i,k}^T A(x_j))}$ is the output of a softmax function, $z_c \in (0, 1)$, $y$ is the one-shot encoded vector, $y_c \in \{0, 1\}$, thus, $|z_c - y_c| > 0$. Finally, we finish the proof:

$$\Delta_{intra} \geq (p \cdot \nabla g_a \cdot \nabla A)^T (x - A(x)), \tag{16}$$

where $p = \sum_{c=1}^{C}(z_c - y_c)$, $\nabla g_a$ is the gradient of the local model $w_i$, and $\nabla A$ is the gradient of the data augmentation function $A$. □

Then we provide the proof the Theorem 3.1 with Lemma A.1 as follows:

*Proof.* Based on Lemma A.1, we derive the intra-model inconsistency for any local models trained by existing OFL methods. When $\nabla g_a \neq 0$ and $\|A(x) - x\| > 0$, $\nabla A \neq 0$, we have:

$$
\begin{aligned}
\|\Delta_{intra}\|^2 &= \|(p \cdot \nabla g_a \cdot \nabla A)^T (x - A(x))\|^2 \\
&= p^2 \cdot \|\nabla g_a\|^2 \cdot \|\nabla A\|^2 \cdot \|(x - A(x))\|^2
\end{aligned}
\tag{17}
$$

Since $|p| > 0$, $\|\nabla g_a\|^2 > 0$, $\|\nabla A\|^2 > 0$, and $\|(x - A(x))\|^2 > 0$, we can induce that $\|\Delta_{intra}\|^2 > 0$.

We finish the proof. □

Note that in Theorem 3.1, the $\nabla g_a = 0$ represents the augmented data $A(x)$ is not sensitive to the local model $w_i$, which is a rare case in practice. If the $\nabla A = 0$, it means the data augmentation function $A$ have fully changed the semantics of the original data $x$. In conclusion, we can infer that the intra-model inconsistency is inevitable in existing OFL methods.

# B. Proof of Theorem 3.2

*Proof.* We derive the gradient of cross-entropy loss as :

$$
\begin{aligned}
\Delta_w &= \sum_c^C \left( \nabla(\Phi_{u,c}\theta_{u,c}) - \nabla(\Phi_{v,c}\theta_{v,c}) \right) \\
&\approx \sum_c^C \left( \frac{\eta n_{u,c}}{n_u}(1 - \overline{z}_{u,c})\overline{x}_{u,c} - \frac{\eta n_{v,c}}{n_v}(1 - \overline{z}_{v,c})\overline{x}_{v,c} \right. \\
&\qquad \left. - \left( \frac{\eta}{n_u} \sum_{c' \in [C_u]\setminus c} n_{u,c'}\overline{z}_{u,c'}\overline{x}_{u,c'} - \frac{\eta}{n_v} \sum_{c' \in [C_v]\setminus c} n_{v,c'}\overline{z}_{v,c'}\overline{x}_{v,c'} \right) \right)
\end{aligned}
\tag{18}
$$

where the term of $\approx$ holds from the Property 1 in (Zhang et al., 2022b). If $n_u = n_v$, we have

$$
\begin{aligned}
\Delta_w = \sum_c^C \Big\{ \frac{\eta}{n_u} \big[ &\underbrace{(n_{u,c}(1 - \overline{z}_{u,c})\overline{x}_{u,c} - n_{v,c}(1 - \overline{z}_{v,c})\overline{x}_{v,c})}_{\Delta_w^+} \\
&\underbrace{- \big( \sum_{c' \in [C_u]\setminus c} n_{u,c'}\overline{z}_{u,c'}\overline{x}_{u,c'} - \sum_{c' \in [C_v]\setminus c} n_{v,c'}\overline{z}_{v,c'}\overline{x}_{v,c'} \big) \big] \Big\}.}_{\Delta_w^-}
\end{aligned}
\tag{19}
$$

We consider the $\|\Delta_w\|^2$ under heterogeneous data distribution, and we omit the $\sum_c^C$ for simplicity.

When $\overline{x}_{u,c} \neq \overline{x}_{v,c}$, we have

$$
\|\Delta_w\|^2 \geq \frac{\eta^2}{n^2} \left| \|\Delta_w^+\|^2 - \|\Delta_w^-\|^2 \right| > 0.
\tag{20}
$$

When $\overline{x}_{u,c} = \overline{x}_{v,c}, c \in [C_u] \cap [C_v], \overline{z}_{u,c} = \overline{z}_{v,c}$, that is, two clients have same prediction and feature on selected positive class $c$, then $c' \in [C_u]\setminus\{[C_u] \cap [C_v]\}$, we have

$$
\begin{aligned}
\|\Delta_w\|^2 &= \frac{\eta^2}{n^2} \|\Delta_w^-\|^2 \\
&= \frac{\eta^2}{n^2} \Big\| \sum_{c' \in [C_u]\setminus\{[C_u]\cap[C_v]\}} n_{u,c'}\overline{z}_{u,c'}\overline{x}_{u,c'} - \sum_{c' \in [C_u]\setminus\{[C_u]\cap[C_v]\}} n_{v,c'}\overline{z}_{v,c'}\overline{x}_{v,c'} \Big\|^2
\end{aligned}
\tag{21}
$$

If we assume the equation above equals to zero, we have $\overline{z}_{u,c'}\overline{x}_{u,c'} = \overline{z}_{v,c'}\overline{x}_{v,c'}$, which requires a perfect feature extractor for all clients. It is not a practical condition in FL according to (Zhou et al., 2024).

When $\overline{x}_{u,c} = \overline{x}_{v,c}, c \in [C]\setminus\{[C_u] \cup [C_v]\}, n_{u,c} = n_{v,c} = 0$, we have the same formulation with Equ. 21, we can obtain $\|\Delta_w\|^2 > 0$.

When $\overline{x}_{u,c} = \overline{x}_{v,c}, c \in [C_u]\setminus\{[C_u] \cap [C_v]\}, n_{v,c} = 0$, we have

$$
\begin{aligned}
\|\Delta_w\|^2 &= \frac{\eta^2}{n^2} \Big\| n_{u,c}(1 - \overline{z}_{u,c})\overline{x}_{u,c} - \big( \sum_{c' \neq c} n_{u,c'}\overline{z}_{u,c'}\overline{x}_{u,c'} - \sum_{c' \neq c} n_{v,c'}\overline{z}_{v,c'}\overline{x}_{v,c'} \big) \Big\|^2 \\
&\approx \frac{\eta^2}{n^2} \| n_{u,c}\overline{x}_{u,c} - \overline{z}_{u,c}\overline{x}_{u,c} \|
\end{aligned}
\tag{22}
$$

where the equality holds if and only if $n_{u,c}(1 - \overline{z}_{u,c})\overline{x}_{u,c} = \Delta_w^-$. In the training phase, it is challenging to maintain this condition between any two clients, so we use the term $\approx$ since $\overline{z}_{u,c'} \ll \overline{z}_{u,c} < 1$. We say that the $\|\Delta_w\|^2$ is more likely to be positive. The other cases, $c \in [C_v]\setminus\{[C_u] \cap [C_v]\}$ can get the same conclusion.

In summary, we conclude that the $\|\Delta_w\|^2 > 0$ under different heterogeneous scenarios. We finish the proof. $\qquad\square$

# C. Algorithm Details

Here, we present the detailed algorithm of FAFI. The FAFI consists of two parts. For the client side, the clients perform self-alignment local training for semantically aligned feature extractors and category-wise distinctive prototypes. Then the server aggregates the local prototypes into a global prototype. By dynamically fusing the feature with more information, FAFI makes predictions by computing the similarity between the global prototypes and the fused feature.

---

**Algorithm 1** FAFI

---

**Input:** Number of clients $m$, number of classes $C$, number of local epochs $E$, learning rate $\eta$, batch size $B$, data heterogeneity $\alpha$.

**Output:** Global prototypes $\mathcal{P}_g$, local models $w_i$, prediction of test set $y_{test}$

  **Client Side:**

  **for** each client $i \in [m]$ in parallel **do**

    Sample local data $\mathcal{D}_i$ with heterogeneity $\alpha$.

    Initialize local feature extractor $\theta_i$ and local prototypes $\mathcal{P}_i$.

    **for** each local epoch $e \in [E]$ **do**

      Sample a mini-batch $\mathcal{B}_i$ from $\mathcal{D}_i$.

      Update local feature extractor $\theta_i$ and local learnable prototypes $\mathcal{P}_i$ with self-alignment local training loss:

$$\mathcal{L} = \mathcal{L}_{ssl}(\theta_i) + \mathcal{L}_{proto}(\mathcal{P}_i, \theta_i)$$

    **end for**

    Send the local feature extractor $\theta_i$ and local learnable prototypes $\mathcal{P}_i$ to the server.

  **end for**

  **Server Side:**

  Global prototypes aggregation $\mathcal{P}_g = \frac{1}{M} \sum_{m=1}^{M} \mathcal{P}_m$

  Informative Feature Fusion

  $\alpha_m = 1 - cos(\mathcal{F}_m, \mathcal{F}_m^{\mathcal{N}(\mu,\sigma)})$

$$\mathcal{F}_{\text{fused}} = \sum_{m}^{M} \frac{\alpha_m}{\sum_{v}^{M} \alpha_v} \mathcal{F}_m$$

  Inference $y_{test} = \arg\max_{c \in [C]} cos(\mathcal{F}_{\text{fused}}, \mathcal{P}_{g,c})$

---

# D. More Related Work

## D.1. Prototype Learning

Prototype refers to the representative feature vector of the instances belonging to a specific class. It is a popular and effective method widely used in various tasks, such as supervised classification tasks and unsupervised learning. Since the prototypes can provide abstract knowledge while preserving data privacy, few works introduce prototypes into federated learning. FedProto (Tan et al., 2022) and FedProc (Mu et al., 2023) aim to achieve a feature-wise alignment with global prototypes. CCVR (Luo et al., 2021) generates virtual features based on an approximated Gaussian Mixture Model (GMM). VHL (Tang et al., 2022) builds a virtual homogeneous dataset for mitigating data heterogeneity. However, existing federated prototype learning methods rely on multi-round interactions between the clients and server, which is not practical in one-shot federated learning.

## D.2. Contrastive Learning

Contrastive learning (CL) is a promising direction in self-supervised learning. Contrastive learning constructs positive and negative pairs for each training instance and designs various loss functions to contrast positiveness against negativeness, such as InfoNCE (Oord et al., 2018), and SupCon (Khosla et al., 2020). One branch of CL focuses on selecting the informative positive pairs and negative pairs (Chongjian et al., 2023). Another branch investigates the semantic structure and involves clustering methods to construct more representative prototypes (Caron et al., 2020; Li et al., 2021a).

*Table 5.* **Impact of different local epochs**: in CIFAR-10, CIFAR-100, and Tiny-ImageNe scenarios with skew ratio $\alpha \in \{0.05, 0.1, 0.3, 0.5\}$.

| Local Epoch $E$ | CIFAR-10 | | | | CIFAR-100 | | | | Tiny-ImageNet | | | |
|---|---|---|---|---|---|---|---|---|---|---|---|---|
| | $\alpha = 0.05$ | $\alpha = 0.1$ | $\alpha = 0.3$ | $\alpha = 0.5$ | $\alpha = 0.05$ | $\alpha = 0.1$ | $\alpha = 0.3$ | $\alpha = 0.5$ | $\alpha = 0.05$ | $\alpha = 0.1$ | $\alpha = 0.3$ | $\alpha = 0.5$ |
| 5 | 42.34 | 45.88 | 54.53 | 63.76 | 10.04 | 13.25 | 18.28 | 22.31 | 10.88 | 13.68 | 20.74 | 23.61 |
| 10 | 52.65 | 58.25 | 66.03 | 73.97 | 12.73 | 20.17 | 27.71 | 32.43 | 15.92 | 21.04 | 29.88 | 34.81 |
| 50 | 57.79 | 63.55 | 72.21 | 77.96 | 15.32 | 25.83 | 33.82 | 39.01 | 20.36 | 35.96 | 36.44 | 41.16 |
| 100 | 62.08 | 68.12 | 76.11 | 82.03 | 24.35 | 31.34 | 41.48 | 46.77 | 23.34 | 36.96 | 40.91 | 46.48 |
| 150 | 69.67 | 72.53 | 81.64 | 86.78 | 29.17 | 43.19 | 49.09 | 59.38 | 31.06 | 40.42 | 46.39 | 52.13 |
| 200 | 71.84 | 77.83 | 84.76 | 88.74 | 31.02 | 45.48 | 56.65 | 61.07 | 36.96 | 43.62 | 53.32 | 56.48 |
| 300 | 72.33 | 77.92 | 85.12 | 89.23 | 31.67 | 46.65 | 57.12 | 62.11 | 37.04 | 43.14 | 53.65 | 57.23 |

*Table 6.* **Impact of Model Architecture:** ResNet-18, ResNet-50, and ViT-base on Tiny-ImageNet.

| Architecture | Accuracy(%) | Memory Cost | Computation Cost |
|---|---|---|---|
| ResNet-18 | 57.23 | 11.2 M | 3.2 min/epoch |
| ResNet-50 | 58.12 | 25.6 M | 4.6 min/epoch |
| ViT-base | 58.09 | 86 M | 5.5 min/epoch |

## E. Experimental Details

**Models and Learning Rate.** Following the advanced OFL works (Zeng et al., 2024), we train ResNet-18 on all datasets. We use the SGD optimizer with a momentum coefficient of 0.9, set the batch size to 256, and make local train $E = 200$ steps. We report the best results by varying the learning rate in $0.05, 0.01, 0.001$. We adopt the temperature of contrastive learning as $\tau = 0.5$, the noise is the standard Gaussian noise $\mathcal{N}(0, 1)$. We use the output of the last layer of ResNet-18 as the feature, and the prototype of each category has the same dimension as the feature. Source code and models are available at `https://github.com/zenghui9977/FAFI_ICML25`.

**Federated Settings.** For OFL methods (including FAFI), we adopt the same settings with (Zeng et al., 2024; Zhang et al., 2022a). We set the number of clients as 5, and the local training epochs of 200. For multi-round FL methods, we select six representative methods, including FedAvg (McMahan et al., 2017), FedProx (Li et al., 2020), SCAFFOLD (Karimireddy et al., 2020), FedCav (Zeng et al., 2021; 2025), FedProto (Tan et al., 2022), FedDC (Gao et al., 2022). To ensure a fair comparison framework, all multi-round FL methods are evaluated under identical experimental conditions as OFL approaches, including matching client numbers, equivalent heterogeneous data distributions, and consistent local learning rates. We try to adopt the same local training epochs as OFL methods do in multi-round FL. However, this configuration unexpectedly led to performance degradation that persisted at levels comparable to O-FedAvg for extended training rounds. To effectively highlight the advantages of FAFI, we consequently adopt the conventional local epoch setting (E=5), a parameter configuration widely validated in FL literature. Under this optimized configuration, our experimental results demonstrate that conventional multi-round FL methods still require over 80 communication rounds to attain accuracy levels equivalent to those achieved by FAFI.

## F. More Experimental Results

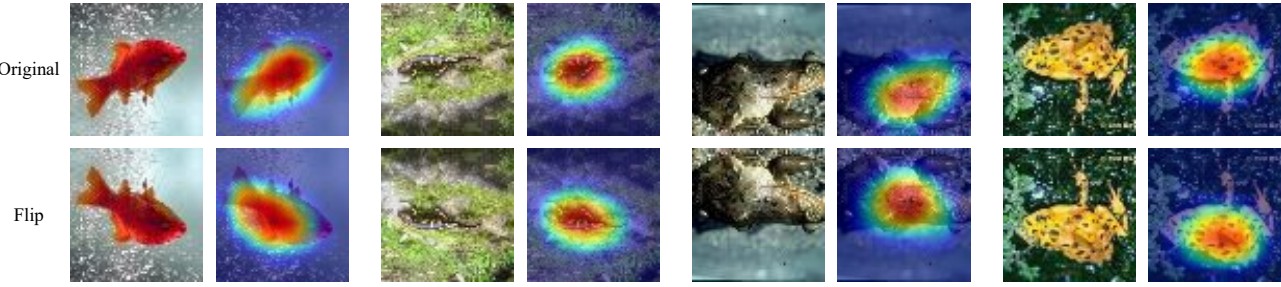

*Figure 6.* **Visualization of FAFI with Grad-Cam.** The features are consistent with the same semantics.

*Table 7.* **Applicability to Resource-constrained Scenarios:** ResNet-18 and MobileNet on CIFAR-10 with $Dir(0.05)$.

| Models | Methods | Batch Size | Memory Cost | CPU Time | GPU Time | Accuracy(%) |
|---|---|---|---|---|---|---|
| MobileNet | O-FedAvg | 512 | 1942 MB | 138 s | 6 s | 10.44 |
| | IntactOFL | 512 | 1942 MB | 141 s | 8 s | 32.31 |
| | FAFI | 32 | 1086 MB | 183 s | 11 s | 55.21 |
| | FAFI | 256 | 1838 MB | 227 s | 14 s | 58.33 |
| | FAFI | 512 | 3082 MB | 240 s | 17 s | 59.64 |
| ResNet-18 | O-FedAvg | 512 | 4638 MB | 197 s | 8 s | 12.13 |
| | IntactOFL | 512 | 4638 MB | 204 s | 10 s | 48.33 |
| | FAFI | 32 | 2789 MB | 307 s | 18 s | 69.73 |
| | FAFI | 256 | 8792 MB | 319 s | 27 s | 70.24 |
| | FAFI | 512 | 14723 MB | 428 s | 32 s | 71.84 |

*Table 8.* **Model Heterogeneity Scenarios:** ResNet-18 and MobileNet on CIFAR-10 with $Dir(0.05)$.

| | Client 0 | Client 1 | Client 2 | Client 3 | Client 4 | IntactOFL | Ours |
|---|---|---|---|---|---|---|---|
| Case #1 | LeNet | ResNet-18 | VGG | MobileNet | ResNet-50 | 48.33 | 71.84 |
| Case #2 | VGG | MobileNet | ResNet-50 | LeNet | ResNet-18 | 52.55 | 72.12 |
| Case #3 | ResNet-50 | LeNet | ResNet-18 | VGG | MobileNet | 54.12 | 72.45 |

*Table 9.* **Impact of the Feature Fusion Strategy in IFFI:** in CIFAR-10 with $Dir(0.05)$.

| Feature Fusion Strategy | CIFAR-10 | CIFAR-100 | Tiny-ImageNet |
|---|---|---|---|
| Average | 72.83 | 37.48 | 30.12 |
| $\mathcal{N}(0,5)$ | 71.83 | 35.4 | 28.09 |
| $\mathcal{N}(0,10)$ | 70.12 | 33.02 | 26.34 |
| $\mathcal{N}(0,50)$ | 68.23 | 31.12 | 24.12 |
| $\mathcal{N}(1,1)$ | 69.12 | 32.77 | 25.12 |
| $\mathcal{N}(0,1)$ (Ours) | 77.83 | 45.48 | 43.62 |

**Impact of Local Epochs.** We tested how different numbers of local training rounds affect performance across three datasets. As shown in Table 5, model accuracy keeps improving when we increase the number of training rounds. However, the improvements become very small after 200 rounds - pushing to 300 rounds only brings less than 0.8% better accuracy compared to 200 rounds, but makes the computation much heavier. After comparing how accuracy improves versus how much computing power is needed, we chose 200 rounds as our standard setting.

**Impact of Model Architecture.** We used ResNet-18 as the default feature extractor in the previous study. In this part, we investigate the impact of different model architectures. We select ResNet-18, ResNet-50, and ViT-base. We compare the memory cost (measured by #parameters), computation cost (measured by the computation time per epoch), and accuracy on Tiny-ImageNet in Table 6. Note that the impact of model architecture on performance is relatively low, but there are significant differences in computational and storage costs. The enhancement in performance is attributed to the designed framework (FAFI), rather than the model.

**Applicability to Resource-constrained Scenarios.** We investigate the applicability of FAFI under the resource-constrained scenarios, which present limited computation capability and memory space. Thus, we select a typical light-weight model, i.e., MobileNet, and report the memory cost and computation cost (CPU or GPU time) in Table 7. We use the GPU memory occupied during local training as the metric for memory cost, while computation cost is measured by the time taken per epoch on the GPU(RTX 4090) and the CPU(Intel Core i7-11700K). We believe that FAFI can achieve competitive performance even in resource-constrained scenarios, requiring only 11 seconds on a GPU with less than 2GB of memory or approximately 3 minutes on a CPU.

**Model Heterogeneity Scenarios.** We test FAFI under model heterogeneity scenarios, where different clients own different model architectures. We list three cases on CIFAR-10 with $Dir(0.1)$ with five different architectures (LeNet, ResNet-18, VGG, MobileNet, ResNet-50). As presented by Table 8, we note that FAFI can still achieve better performance under

heterogeneous models.

**Impact of the Feature Fusion Strategy.** We investigate the impact of different feature fusion strategies in IFFI on CIFAR-10 with $Dir(0, 1)$. We notice that the standard Gaussian distribution achieves better performance than other settings. We attribute this to the fact that the distance from standard Gaussian noise better reflecting the informativeness of the extracted features (Elbatel et al., 2024).

**Visualization.** We visualize the features extracted by the local models trained by FAFI in Figure 6. We select one of the local models trained by FAFI and choose some samples for visualization. We found that the features extracted under the same semantics are consistent, which proves the effectiveness of the Self-alignment Local Training.

