# OpenReview forum: "Does One-shot Give the Best Shot? Mitigating Model Inconsistency in One-shot Federated Learning"
_ICML.cc/2025/Conference — ICML 2025 poster_

### Official Review · Reviewer_8kHb · 2025-02-23

**Overall Recommendation:** 3

**Summary:**

The paper investigates one-shot Federated Learning (OFL), which aims to reduce the communication costs associated with traditional multi-round Federated Learning. The authors highlight that existing OFL methods face significant challenges due to "garbage in, garbage out" issues, where inconsistent local models lead to a degraded global model. The proposed solution, denoted as FAFI, introduces Self-Alignment Local Training (SALT) and Informative Feature Fused Inference (IFFI) to create consistent features in one-shot local models and enhance global model inference. Through extensive experiments, FAFI outperforms existing OFL methods by a significant margin.

**Claims And Evidence:**

Claims made in the submission are supported by clear and convincing evidence.

**Essential References Not Discussed:**

For the feature alignment, there are some works also enhance the feature quality including [1,2,3], in which aggregating same-class features and repelling features of different classes are also proposed. For prototype based FL, works [1,4,5] are also related. Authors should consider adding them for discussion, what differences of the proposed feature alignment and prototype learning from them. Moreover, in Section 4.3, what is the difference between the proposed feature fusion and FuseFL?

[1] Virtual Homogeneity Learning: Defending against Data Heterogeneity in Federated Learning. In ICML 2022.
[2] Model-Contrastive Federated Learning. In CVPR 2021.
[3] FedImpro: Measuring and Improving Client Update in Federated Learning. In ICLR 2024.
[4] Fedproto: Federated prototype learning across heterogeneous clients. In AAAI 2022.
[5] No fear of heterogeneity: Classifier calibration for federated learning with non-iid data. In NeurIPS 2021.

**Experimental Designs Or Analyses:**

The experimental design is comprehensive. However, the feature fusion part might introduce extra communication or computation, authors can  provide more detailed interpretation of this.

**Methods And Evaluation Criteria:**

The proposed methods and evaluation criteria (e.g., benchmark datasets) make sense for the problem at hand.

**Other Comments Or Suggestions:**

See weaknesses.

**Other Strengths And Weaknesses:**

Strengths:
1. The paper thoroughly identifies and analyzes the inconsistencies within and between one-shot local models, providing a theoretical foundation for the proposed method.
2. The introduction of Self-Alignment Local Training (SALT) effectively addresses intra-model inconsistencies by fostering invariant feature representations.
3. The use of Informative Feature Fused Inference (IFFI) on the server-side improves the aggregation process, leading to better global model performance.
4. Extensive empirical validation across multiple datasets demonstrates the effectiveness of the FAFI framework in a variety of scenarios, establishing its robustness.
5. The proposed solution achieves a notable accuracy improvement than baseline models, showcasing its potential impact on the field.


Weaknesses:
1. The paper does not address the potential limitations and complexities of implementing the FAFI framework in real-world federated learning environments.
3. There may be a lack of comprehensive comparisons with other state-of-the-art methods that also aim to reduce communication costs, apart from the eleven baselines included.
4. Although the paper discusses privacy concerns, the utilized label information may still expose data privacy at some degree.
5. For the feature alignment, there are some works also enhance the feature quality including [1,2,3], in which aggregating same-class features and repelling features of different classes are also proposed. For prototype based FL, works [1,4,5] are also related. Authors should consider adding them for discussion, what differences of the proposed feature alignment and prototype learning from them. Moreover, in Section 4.3, what is the difference between the proposed feature fusion and FuseFL?
6. The paper could benefit from a more in-depth analysis of the computational efficiency and resource requirements of the FAFI framework compared to existing methods.
7. Some format errors. Line 216, the formula is out of box.
8. Some writing issues. Line 164, the \Delta_intra is not defined in the main texts. The augmentation function A is not clear, why we need an augmentation function A here. I guess the meaning of A in appendix is the global dataset.


[1] Virtual Homogeneity Learning: Defending against Data Heterogeneity in Federated Learning. In ICML 2022.
[2] Model-Contrastive Federated Learning. In CVPR 2021.
[3] FedImpro: Measuring and Improving Client Update in Federated Learning. In ICLR 2024.
[4] Fedproto: Federated prototype learning across heterogeneous clients. In AAAI 2022.
[5] No fear of heterogeneity: Classifier calibration for federated learning with non-iid data. In NeurIPS 2021.

**Questions For Authors:**

See weaknesses.
1. Besides, to highlight the novelty, could you illustrate the detailed differences between the proposed methods with previous works?
2. In Line 69, ``we observe that better performance is always accompanied by larger parameter discrepancies.''. However, later analysis shows that the large discrepancy is not good so you want to minimize (Line 258). So, could you interpret this with more details?
I'm willing to increase my score if addressing above weaknesses and questions.

**Relation To Broader Scientific Literature:**

This work is related to the one-shot Federated learning. The experiment results reveals general weakness of previous OFL methods.
The Figure 4 also provides a holistic view to compare accuracy and communication costs of different OFL and multi-round FL methods.

**Theoretical Claims:**

I checked the correctness of proofs for theoretical claims.

---

> ### Author Rebuttal · Authors · 2025-03-31
>
> We would like to thank the reviewer for the time in reviewing. We appreciate that you consider our presentation is clear, and our method is effective and robust. Please see our detailed feedback for your concerns below.
>
> **W1: Potential solutions and complexities of implementation.**
>
> **Ans for W1:** Thank you for your valuable comments. Indeed, FAFI may have limitations in handling domain shifts/multi-task scenarios. FAFI assumes that invariant feature representations should remain consistent or similar, enabling them to mitigate model inconsistencies. However, in domain shift/multi-task scenarios, feature representations often exhibit significant discrepancies across different domains or tasks, particularly in the isolated training paradigm.
>
> A potential solution is to cluster and ensemble feature representations across various domains or tasks, allowing unbiased representations to encapsulate multi-domain/task knowledge.
>
> Regarding implementation in real-world FL environments, as detailed in Algorithm.1(Appendix C), FAFI requires only two modifications compared to existing OFL methods, making it easily adaptable for both clients and the server:
> 1) On the client side, we train the feature extractor in a contrastive manner and use the prototypes instead of the classifier.
> 2) On the server side, fusion is performed at the feature level rather than the parameter or prediction level.
>
> We will highlight these in our revised version.
>
> **W2: Comparisons with other methods**
>
> **Ans for W2:** Thanks for your important comments. We have conducted new experiments comparing 3 methods, i.e., FedCompress, FedKD, and FedACG, which reduce the communication cost by gradient compression, gradient fractorization, and transferring gradient momentum. These new results on CIFAR-10 are provided below and added to our revision. Results show that FAFI achieves better efficiency compared with all baselines.
>
> |Methods|Acc.|Comm. Cost|
> |-|-|-|
> |FedCompress(C=1)|15.45|9.86MB|
> |FedKD(C=1)|20.67|34.89MB|
> |FedACG(C=1)|19.77|44.7MB|
> |FedCompress(C=50)|33.34|0.48GB|
> |FedKD(C=50)|60.12|1.70GB|
> |FedACG(C=50)|68.23|2.18GB|
> |Ours|77.83|44.7MB|
>
> **W3: Privacy issue**
>
> **Ans for W3:** Thanks for your very valuable comments. We agree with the statement *there is no free lunch for the privacy-utility trade-off*[1], for that the tenet of FL is to seek balance among efficiency, effectiveness, and privacy. FAFI, as well as most OFL methods, all aim to ensure efficiency for affordable deployment while striving to find the effectiveness boundaries as much as possible and maintain privacy below an acceptable level. As to FAFI, the possible leakage is the prototypes, which have been proven in previous works to compromise no privacy. For stricter privacy requirements, one potential solution is implementing differential privacy or using noised samples with the cost of some effectiveness degradation.
>
> **W4 & Q1: Comparisons with other methods**
>
> **Ans for W4 & Q1:** Thanks for your valuable comments. We present the difference between our proposed FAFI with feature-enhanced and prototype-based methods as follows:
>
> For feature-enhanced works, we note that FAFI is one-shot and selective enhances the feature at the inference stage for high performance.
>
> |Methods|Paradigm|Stage|Feature Enhancemnet|
> |-|-|-|-|
> |VHL|Mul.|Train|Generative|
> |MOON|Mul.|Train|Constrastive|
> |FedImpro|Mul.|Train|Estimative|
> |FuseFL|One-shot|Train|Adaptor-based|
> |**Ours**|**One-shot**|**Inference**|**Selective**|
>
> For prototype-based methods, FAFI aligns global prototypes by aggregating learnable local prototypes in one round.
>
> |Methods|Paradigm|Prototypes|
> |-|-|-|
> |VHL|Mul.|Generative|
> |FedProto|Mul.|Statistical|
> |CCVR|Mul.|GMM-based|
> |**Ours**|**One-shot**|**Learnable**|
>
> Mul. is short for multi-round. We will add these discussions in our revision.
>
> **W5: More in-depth efficiency analysis**
>
> **Ans for W5:** Thanks for your valuable comments. We have conducted the memory and computation cost analysis of FAFI. Due to limited space, please see the response to **Reviewer Chua W1 & Q1**.
>
> **W6 & W7 & Q2: Writing issues**
>
> **Ans:** Thanks for your careful reading.
> 1) The $\Delta_{intra} = \vert L(x, y) - L(x', y) \vert$ is the performance discrepencies between any two samples $x, x'$ with the same label $y$. For any two samples $x, x'$, we can find a function $A$ that satisfies $x' = A(x)$. $A$ can abstract the feature variation among samples.
> 2) The 'performance' in L69 represents the performance of the **local model**, we observe that better local performance is always accompanied by large parameter discrepancies(Fig.2-b). The performance in L258 denotes the performance of the **global model**.
>
> Sorry for these misunderstandings. We will revise these writing issues and format errors and provide more explanations for clarity.
>
> [1] No free lunch theorem for security and utility in federated learning. ACM TIST, 2022.

---

> > ### Comment · Reviewer_8kHb · 2025-04-02
> >
> > Thanks for the responses. The provided new clarification is clear, and the differences from previous methods are clear. I'd like to raise my score.

---

> > > ### Author Response · Authors · 2025-04-02
> > >
> > > Dear Reviewer 8kHb:
> > >
> > > We're grateful for your quick feedback during this busy period. We deeply appreciate your consideration in raising the score.  Your constructive comments have significantly contributed to the refinement of our work. We will add all these above discussions in our final revision. Thanks a lot for your valuable comments!
> > >
> > > We will remain open and ready to delve into any more questions or suggestions you might have until the last moment.
> > >
> > > Best regards and thanks

---

### Official Review · Reviewer_BvHZ · 2025-03-03

**Overall Recommendation:** 4

**Summary:**

Existing OFL methods focus on server-side aggregation, which falls into ‘garbage in garbage out’ pitfall. They unravel the root cause of such garbage inputs as intra-model and inter-model inconsistencies in the face of data heterogeneity. To address these, they design self-alignment local training and informative feature fused inference. Experimental results verify the effectiveness, scalability, and efficiency of the proposed method.

**Claims And Evidence:**

Clear and convincing claims.

**Essential References Not Discussed:**

Almost all essential references have been included and discussed.

**Experimental Designs Or Analyses:**

I have assessed the soundness and validity of the experimental designs, and the authors have effectively validated the proposed method’s effectiveness.

**Methods And Evaluation Criteria:**

The method is suitable for the proposed model inconsistency problem. The evaluation criteria, accuracy on three classification datasets, is suitable.

**Other Comments Or Suggestions:**

1) Inconsistent descriptions of existing OFL methods categories (Sec. 2 and Sec. 5)

**Other Strengths And Weaknesses:**

1. Strengths:
1) This manuscript proposes FAFI which achieves a good performance while requiring only one communication round.
2) The paper is the first to systematically identify the "garbage in, garbage out" pitfall in OFL caused by intra- and inter-model inconsistencies. The visualization and theoretical analysis on the model inconsistencies are interesting.
3) The authors utilize self-supervised learning methods and prototype learning for better local models and propose a feature fusion-based inference method. The feature fusion is novel in OFL.
4) FAFI is data-free, which does not require the source data or any other auxiliary information.
2. Weaknesses:
1) SALT consists of two parts, feature alignment, and category-wise prototype learning, and how these two parts contribute to mitigating the intra-model inconsistency. Please provide the ablation study on L_ssl and L_proto.
2) Noise impact in IFFI. How does the noise representation used for attention weighting in feature fusion (§4.3) impact the final performance. Please provide more evaluations to demonstrate the impact of the noise.

**Questions For Authors:**

1. It is unclear how L_ssl and L_proto contribute to mitigating the intra-model inconsistency
2. How does the weights in feature fusion impact the performance.

**Relation To Broader Scientific Literature:**

They provide a novel solution for mitigating model inconsistency. Existing OFL methods all focus on server-side aggregation based on inferior local models. This paper proposes to provide a high-quality local model through self-alignment local training, and utilizing the extracted feature for aggregation.

**Theoretical Claims:**

Correct theoretical claims. The manuscript contains two theorems, Theorem 3.1 for intra-model inconsistency and Theorem 3.2 for inter-model inconsistency.

---

> ### Author Rebuttal · Authors · 2025-03-30
>
> We sincerely thank the reviewer for taking the time to review our work. We greatly appreciate you find our method is efficient and data-free. Please find our detailed responses to your concerns below.
>
>
> **W1 & Q1: Ablation study on $L_{ssl}$ and $L_{proto}$.**
>
> **Ans for W1 & Q1:** Thanks for your important comments. As suggested, we have conducted new ablation studies on $L_{ssl}$ and $L_{proto}$, and the results are shown below.
>
>
> | Feature Extractor | Prototypes / Classifier | CIFAR-10 | CIFAR-100 | Tiny-ImageNet |
> | ----------------- | ---------------------- | -------- | ---------- | ------------- |
> | $L_{ce}$ | Classifier + $L_{ce}$ |  17.34 | 6.45 | 8.31 |
> | $L_{ce}$ | Prototypes + $L_{ce}$ |  18.23 | 7.12 | 8.92 |
> | $L_{ce}$ | Prototypes + $L_{proto}$ |  52.34 | 33.34 | 23.12 |
> | $L_{ssl}$ | Classifier + $L_{ce}$ |  22.34 | 12.45 | 10.44 |
> | $L_{ssl}$ | Prototypes + $L_{ce}$ |  50.12 | 31.89 | 22.34 |
> | $L_{ssl}$ | Prototypes + $L_{proto}$ |  77.83 | 45.48 | 43.62 |
>
> We note that the performance of FAFI is significantly improved by using $L_{ssl}$ and $L_{proto}$ for enhanced feature representation and discriminative prototypes, and the combination of $L_{ssl}$ and $L_{proto}$ achieves the best. Additional details will be added in Appendix F in our revision.
>
>
> **W2 & Q2: Impact of the feature fusion strategy.**
>
> **Ans for W2 & Q2:** Thanks for your important comments. As suggested, we have conducted new evaluations on the impact of the feature fusion strategy with $Dir(0.1)$, and the results are shown below.
>
> | Feature Fusion Strategy | CIFAR-10 | CIFAR-100 | Tiny-ImageNet |
> | -------- | -------- | -------- | -------- |
> | Average | 72.83 | 37.48 | 30.12 |
> | $\mathcal{N}(0,5)$ | 71.83 | 35.40 | 28.09 |
> | $\mathcal{N}(0,10)$ | 70.12 | 33.02 | 26.34 |
> | $\mathcal{N}(0,50)$ | 68.23 | 31.12 | 24.12 |
> | $\mathcal{N}(1,1)$ | 69.12 | 32.77 | 25.12 |
> | $\mathcal{N}(0,1)$ (Ours) | 77.83 | 45.48 | 43.62 |
>
> We note that $\mathcal{N}(0,1)$ facilitates the best performance in the tested cases, as it effectively captures feature information compared to other strategies. Additional details and discussions about this hyperparameter will be included in Appendix F in the revision.

---

> > ### Comment · Reviewer_BvHZ · 2025-04-06
> >
> > Thank you for the author's response. After reading the author's rebuttal, the main concerns I had have been addressed, and I will maintain my score.

---

> > > ### Author Response · Authors · 2025-04-06
> > >
> > > Dear Reviewer BvHZ:
> > >
> > > We're grateful for your quick feedback during this busy period. We deeply appreciate your consideration in raising the score. Your constructive comments have significantly contributed to the refinement of our work. Thanks a lot for your valuable comments! We will remain open and ready to delve into any more questions or suggestions you might have until the last moment.
> > >
> > > Best regards and thanks

---

### Official Review · Reviewer_WoAM · 2025-03-09

**Overall Recommendation:** 4

**Summary:**

This manuscript tries to solve the 'garbage in garbage out' problem caused by inconsistent models in the one-shot federated learning paradigm. They propose FAFI, a novel OFL framework consisting of two key components: SALT for invariant feature learning and IFFI for server-side feature fusion-based inference. Extensive experiments on CIFAR-10/100 and Tiny-ImageNet demonstrate FAFI’s superiority over 11 baselines, achieving a 10.86% average accuracy improvement.

**Claims And Evidence:**

The claims made in this manuscript are supported by clear and convincing evidence.

**Essential References Not Discussed:**

The paper has cited the related works that are essential to understanding the key contribution.

**Experimental Designs Or Analyses:**

The experimental parts are well-organized, the demonstrations on main paper and appendix are good. Better presenting more results on efficiency.

**Methods And Evaluation Criteria:**

The proposed method and evaluation criteria are suitable for the problem.

**Other Comments Or Suggestions:**

1. Some typos in appendix, e.g., line 653 $Vert(x-A(x))||^2>0$ should be $|| x - A(x) ||^2>0$.
2. It is recommended that open-source code be provided to replicate the experimental results.

**Other Strengths And Weaknesses:**

Strengths:
1. The paper is well-demonstrated and well-organized. The figures, such as Figure 1(a), Figure 2(a), and Figure 3 are good.
2. The paper is well-motivated. The analysis of intra- and inter-model inconsistencies is supported by both empirical evidence (Grad-CAM visualizations) and theoretical proofs (Theorems 3.1 and 3.2).
3. The experimental part is well-organized. Evaluations across diverse non-IID settings and clients scales over 11 baselines validate FAFI's effectiveness, scalability, and efficiency.
4. The method significantly reduces communication overhead while outperforming multi-round FL baselines in efficiency-accuracy trade-offs (Figure 4). This aligns well with real-world FL deployment needs.

Weaknesses:
1. Details on the noise in IFFI (eq. 8) are unclear. It is better to provide code and hyperparameter settings.
2. The efficiency on extreme non-IID settings, such as $\alpha=0.05$. Figure 4 only provides the evaluations on $\alpha=0.5$. Please provide more settings to verify its efficiency.

**Questions For Authors:**

1. Please provide the details about the noise in IFFI.
2. Please provide more settings to verify the efficiency of FAFI.

**Relation To Broader Scientific Literature:**

The paper situates its contributions within the broader federated learning (FL) literature by empirically and theoretically analyzing the limitations of existing one-shot FL (OFL) methods. The paper explicitly differentiates FAFI from related approaches in Sec.4.4, including prototype-based methods and model merging techniques.

**Theoretical Claims:**

The theoretical claims in this paper are correct. However, some typos in appendix, line 653 $Vert(x-A(x))||^2>0$ should be $|| x - A(x) ||^2>0$ ?

---

> ### Author Rebuttal · Authors · 2025-03-30
>
> We sincerely thank the reviewer for taking the time to review our work. We greatly appreciate you find the paper is well-motivated and well-demonstrated. Please find our detailed responses to your concerns below.
>
> **W1 & Q1: Details in IFFI.**
>
> **Ans for W1 & Q1**: Thanks for your important comments. Sorry for missing the explanations about IFFI. We use the Gaussian distribution $\mathcal{N}(0, 1)$ in Equ.8. For clarity, we will add an ablation study on the hyperparameters. More details can be seen in response to **Reviewer BvHZ W2 & Q2** and will be added in Appendix E in the revision.
>
>
> **W2 & Q2: More settings to verify the effectiveness of FAFI.**
>
> **Ans for W2 & Q2:** Thanks for your important comments. We have conducted new evaluations on CIFAR-10 with data heterogeneity $Dir(0.05)$ and $Dir(0.1)$, and the results are shown below.
>
> | Methods |  Acc. $Dir(0.05)$  | Acc. $Dir(0.1)$ | Comm. Cost |
> | ------------- | - | - | - |
> | MA-Echo | 36.77 | 51.23 | 44.7 MB  |
> | O-FedAvg | 12.13 | 17.43 | 44.7 MB  |
> | FedFisher | 40.03 | 47.01 | 48.2 MB |
> | FedDF | 35.53 | 41.58 | 44.7 MB |
> | F-ADI | 35.93 | 48.35 | 44.7 MB |
> | F-DAFL | 38.32 | 46.34 | 44.7 MB |
> | DENSE | 38.37 | 50.26 | 44.7 MB |
> | Ensemble | 41.36 | 45.43 | 44.7 MB |
> | Co-Boosting | 39.20 | 58.49 | 44.8 MB |
> | FuseFL | 54.42 |  73.79 | 53.32 MB |
> | IntactOFL | 48.22 | 61.13 | 44.7 MB |
> | FedAvg $C=50$ | 23.45 | 27.44 | 2.12 GB |
> | FedProx $C=1$ | 13.53 | 17.58 | 44.7 MB |
> | FedProx $C=50$ | 23.32 | 26.76 | 2.12 GB |
> | SCAFFOLD $C=1$ | 12.45 | 16.23 | 89.4 MB |
> | SCAFFOLD $C=50$ | 27.22 | 30.45 | 4.36 GB |
> | FedCav $C=1$ | 12.49 | 16.77 | 44.8 MB |
> | FedCav $C=50$ | 26.45 | 30.23 | 2.12 GB |
> | FedProto $C=1$ | 12.11 | 16.23 | 44.7 MB |
> | FedProto $C=50$ | 28.31 | 32.55 | 2.12 GB |
> | FedDC $C=1$ | 11.32 | 15.23 | 44.7 MB |
> | FedDC $C=50$ | 30.23 | 44.23 | 2.12 GB |
> | Ours | 71.84 | 77.83 | 44.7 MB |
>
> $C$ is the communication rounds. **We note that FAFI can achieve competitive performance even in extreme heterogeneous scenarios (i.e., $Dir(0.05)$)**, requiring only 44.7 MB of communication cost. Additional details will be included in Appendix F in our revised version.
>
> **C1 & C2: Typos and open source code**
>
> **Ans for C1 & C2**: Thanks for your comments. We will fix the typos in Appendix A, and we promise we will open the source code.

---

### Official Review · Reviewer_Chua · 2025-03-11

**Overall Recommendation:** 4

**Summary:**

The paper addresses the critical challenge of model inconsistency in OFL due to heterogeneous data. They identify two key inconsistencies—intra-model and inter-model—and propose a novel framework, FAFI, which combines client-side self-aligned training (SALT) and server-side informative feature fusion (IFFI). Extensive experiments on three classification datasets demonstrate significant performance improvements over 11 baselines.

**Claims And Evidence:**

The claims are clear and convincing.

**Essential References Not Discussed:**

References are essential and sufficient for understanding the key contributions.

**Experimental Designs Or Analyses:**

I have checked experimental designs or analyses.

**Methods And Evaluation Criteria:**

FAFI consists of SALT and IFFI for intra- and inter-model inconsistencies, respectively. Evaluations on CIFAR-10/100 and Tiny-ImageNet are appropriate.

**Other Comments Or Suggestions:**

No

**Other Strengths And Weaknesses:**

Strengths:
- The proposed method is technically sound. Moreover, leveraging self-supervised methods and prototype learning is novel in OFL scenarios
- Comprehensive survey on existing OFL methods and good discussion on analogous methods, such as prototype-based FL and model merging approaches.
- Significant performance improvement and extensive and sufficient evaluations.
Weaknesses:
- Computational costs for SALT’s contrastive learning (e.g., batch size=256) seem to be expensive for resource-constrained clients, which could be limited in edge scenarios with restricted computational capabilities.
- The reliance on class-wise prototypes assumes a fixed and known number of classes, which may not hold in dynamic or open-set FL scenarios.
- Some notions in theoretical analysis lacks of explanations.
- Federated learning under heterogeneous model scenarios is a very practical research problem, the model architectures across different clients are different. FAFI only consider the homogeneous model scenarios.

**Questions For Authors:**

- My main concern with this work is its applicability to resource-constrained scenarios, such as edge or mobile environments. In these scenarios, they tend to use more lightweight models~(MobileNet). Can FAFI perform well on some lightweight models? Besides, the diverse computation capabilities across clients restrict the clients from adopting the models with the same architecture, Can FAFI support heterogeneous model architectures?

**Relation To Broader Scientific Literature:**

Unlike approaches that focus on server-side aggregation, this paper emphasizes improving local training strategies to achieve better models and mitigate the ‘garbage in, garbage out’ pitfall.

**Theoretical Claims:**

I have checked the theoretical claims and proofs. Theorem 3.1 for intra-model inconsistency, Theorem 3.2 for inter-model inconsistency, and their proofs in the appendix are correct.

---

> ### Author Rebuttal · Authors · 2025-03-30
>
> We sincerely thank the reviewer for taking the time to review our work. We greatly appreciate your recognition of the proposed method as both technically sound and high-performing. Please find our detailed responses to your concerns below.
>
> **W1 & Q1: Applicability to resource-constrained scenarios.**
>
> **Ans for W1 & Q1:** Thanks for your important comments. We note that **our proposed FAFI can support lightweight models and is applicable to resource-constrained scenarios**. We have complemented statistics on computational and memory cost with MobileNet and ResNet-18 on CIFAR-10.
>
> | Methods            | Memory Cost | Computation Cost (GPU/ CPU)  | Accuracy |
> |-|-|-|-|
> |MobileNet|
> | O-FedAvg | 1942 MB | 6s / 138s | 10.44 |
> | IntactOFL | 1942 MB | 8s / 141s | 32.31 |
> | Ours $b$=32 | 1086 MB | 11s / 183s | 55.21 |
> | Ours $b$=256 | 1838 MB | 14s / 227s | 58.33 |
> | Ours $b$=512 | 3082 MB | 17s / 240s | 59.64 |
> | ResNet-18 |
> | O-FedAvg | 4638 MB | 8s / 197s | 12.13 |
> | IntactOFL | 4638 MB | 10s / 204s | 48.33 |
> | Ours $b$=32 | 2789 MB | 18s/ 307s | 69.73 |
> | Ours $b$=256 | 8792 MB | 27s / 319s | 70.24 |
> | Ours $b$=512 | 14723 MB | 32s / 428s | 71.84 |
>
> We use the GPU memory occupied during local training as the metric for memory cost, while computation cost is measured by the time taken per epoch on GPU(RTX 4090) and CPU(Intel Core i7-11700K). Here, $b$ represents the batch size. We believe that FAFI can achieve competitive performance even in resource-constrained scenarios, requiring only 11 seconds on a GPU with less than 2GB of memory or approximately 3 minutes on a CPU. Additional details will be included in our revised version.
>
> **W2: Assumption on the number of classes, which may not hold in dynamic or open-set FL scenarios.**
>
> **Ans for W2:** Thanks for your comments. FAFI assumes that the number of classes is known in advance. However, in dynamic or open-set FL scenarios, the number of classes may change over time. We note that FAFI can be easily extended to these scenarios by extracting the invariant features of the new classes and learning a new discriminative prototype for them. We will include this discussion in our revision.
>
> **W3: Some notions are not clear.**
>
> **Ans for W3:** Thanks for your comments. Sorry for the unclear notations. For clarity, 1) the $\Delta_{intra} = \vert L(x, y) - L(x', y) \vert$ is the performance discrepancies between any two samples $x, x'$ with the same label $y$; 2) $A$ refers to the function that can abstract the feature variation among samples. We will clarify these notations in the revision.
>
> **W4 & Q1: Model heterogeneity scenarios.**
>
> **Ans for W4 & Q1:** Thanks for your comments. We note that FAFI can support heterogeneous models. We have conducted new evaluations on heterogeneous models with five different architectures (LeNet, ResNet-18, VGG, MobileNet, ResNet-50) on CIFAR-10 with data heterogeneity $Dir(0.1)$ as follows and added more details in Appendix F. We only report part of them here due to the limited space.
>
> | Client 0 | Client 1 | Client 2 | Client 3 | Client 4 | IntactOFL | Ours |
> | -------- | -------- | ------- | ------- | ------- | --------- | ---- |
> | LeNet | ResNet-18 | VGG | MobileNet | ResNet-50 | 48.33 | 71.84 |
> | VGG | MobileNet | ResNet-50 | LeNet | ResNet-18 | 52.55 | 72.12 |
> | ResNet-50 | LeNet | ResNet-18 | VGG | MobileNet | 54.12 | 72.45 |

---

> > ### Comment · Reviewer_Chua · 2025-04-03
> >
> > Thank the authors for further experimental analysis and clarifying the rationale, which have addressed most of my questions and concerns. I appreciate this manuscript's interesting task, clear presentation, and extensive experiments.

---

> > > ### Author Response · Authors · 2025-04-04
> > >
> > > Dear Reviewer Chua:
> > >
> > > We're grateful for your quick feedback during this busy period. We deeply appreciate your consideration in raising the score.  Your constructive comments have significantly contributed to the refinement of our work. Thanks a lot for your valuable comments! We will remain open and ready to delve into any more questions or suggestions you might have until the last moment.
> > >
> > > Best regards and thanks

---

### Decision · Program_Chairs · 2025-05-01

**Decision:**

Accept (poster)

**Comment:**

The paper addresses the OFL’s challenges due to the inconsistent models. Two technical innovations are proposed and achieved over 10% model accuracy improvement.